# Structure of the PAPP-A_BP5 complex reveals mechanism of substrate recognition

Russell A. Judge [1,7], Janani Sridar[2,7], Kathryn Tunyasuvunakool [3,7], Rinku Jain[1], John C. K. Wang[2], Christna Ouch[4], Jun Xu[2], Amirhossein Mafi[2], Aaron H. Nile[2], Clint Remarcik[2], Corey L. Smith[5], Crystal Ghosh[2], Chen Xu[4], Vincent Stoll[1], John Jumper [3], Amoolya H. Singh[2,6], Dan Eaton [2] ✉ & Qi Hao [2] ✉

Insulin-like growth factor (IGF) signaling is highly conserved and tightly regulated by proteases including Pregnancy-Associated Plasma Protein A (PAPP-A). PAPP-A and its paralog PAPP-A2 are metalloproteases that mediate IGF bioavailability through cleavage of IGF binding proteins (IGFBPs). Here, we present single-particle cryo-EM structures of the catalytically inactive mutant PAPP-A (E483A) in complex with a peptide from its substrate IGFBP5 (PAPP-A_BP5) and also in its substrate-free form, by leveraging the power of AlphaFold to generate a high quality predicted model as a starting template. We show that PAPP-A is a flexible *trans*-dimer that binds IGFBP5 via a 25-amino acid anchor peptide which extends into the metalloprotease active site. This unique IGFBP5 anchor peptide that mediates the specific PAPP-A-IGFBP5 interaction is not found in other PAPP-A substrates. Additionally, we illustrate the critical role of the PAPP-A central domain as it mediates both IGFBP5 recognition and *trans*-dimerization. We further demonstrate that PAPP-A *trans*-dimer formation and distal inter-domain interactions are both required for efficient proteolysis of IGFBP4, but dispensable for IGFBP5 cleavage. Together the structural and biochemical studies reveal the mechanism of PAPP-A substrate binding and selectivity.

Insulin-like growth factors (IGFs), including IGF1 and IGF2, play important roles in growth, development, metabolism, and aging[1–3]. Dysregulation of IGF levels results in pathologies such as diabetes, chronic liver disease, neurodegeneration, cardiovascular diseases, and other aging related diseases[4,5]. The IGF signaling pathway is therefore a tightly regulated network in which circulating and tissue-associated IGFs are sequestered by insulin-like growth factor binding proteins (IGFBPs), consisting of six family members (IGFBP1-6). The proteolytic cleavage of IGFBPs is required for the release of sequestered IGFs to enable their binding to insulin-like growth factor receptors (IGFRs) for downstream signaling[6–8]. Release of IGFs in humans is mediated by

proteases including two metalloproteases: pregnancy-associated plasma protein-A (PAPP-A, also called pappalysin-1) and its paralog PAPP-A2 (Supplementary Fig. 1)[6,9–13]. Secreted PAPP-A is a 1547-residue, multidomain, dimeric glycoprotein which recognizes and cleaves IGFBP2, 4 and 5[10–17], while PAPP-A2 mainly cleaves IGFBP3 and 5[9,17–21] (Fig. 1a, Supplementary Fig. 1)[11,13,14]. While circulating levels of IGF are regulated by PAPP-A, PAPP-A activity itself is regulated by proMBP, STC1 and STC2 where proMBP and STC2 are reported to inhibit PAPP-A by covalent association[10,11,22–31].

Increased circulating levels of active PAPP-A have been associated with multiple diseases including atherosclerosis, diabetic nephropathy

[1]AbbVie, 1 North Waukegan Road, North Chicago, IL, USA. [2]Calico Life Sciences LLC, South San Francisco, CA, USA. [3]DeepMind, London, UK. [4]Department of Biochemistry & Molecular Biotechnology, University of Massachusetts Chan Medical School, Worcester, MA, USA. [5]AbbVie Bioresearch Center, Worcester, MA, USA. [6]Present address: GRAIL, Menlo Park, CA, USA. [7]These authors contributed equally: Russell A. Judge, Janani Sridar, Kathryn Tunyasuvunakool. ✉e-mail: deaton@calicolabs.com; qhao@calicolabs.com

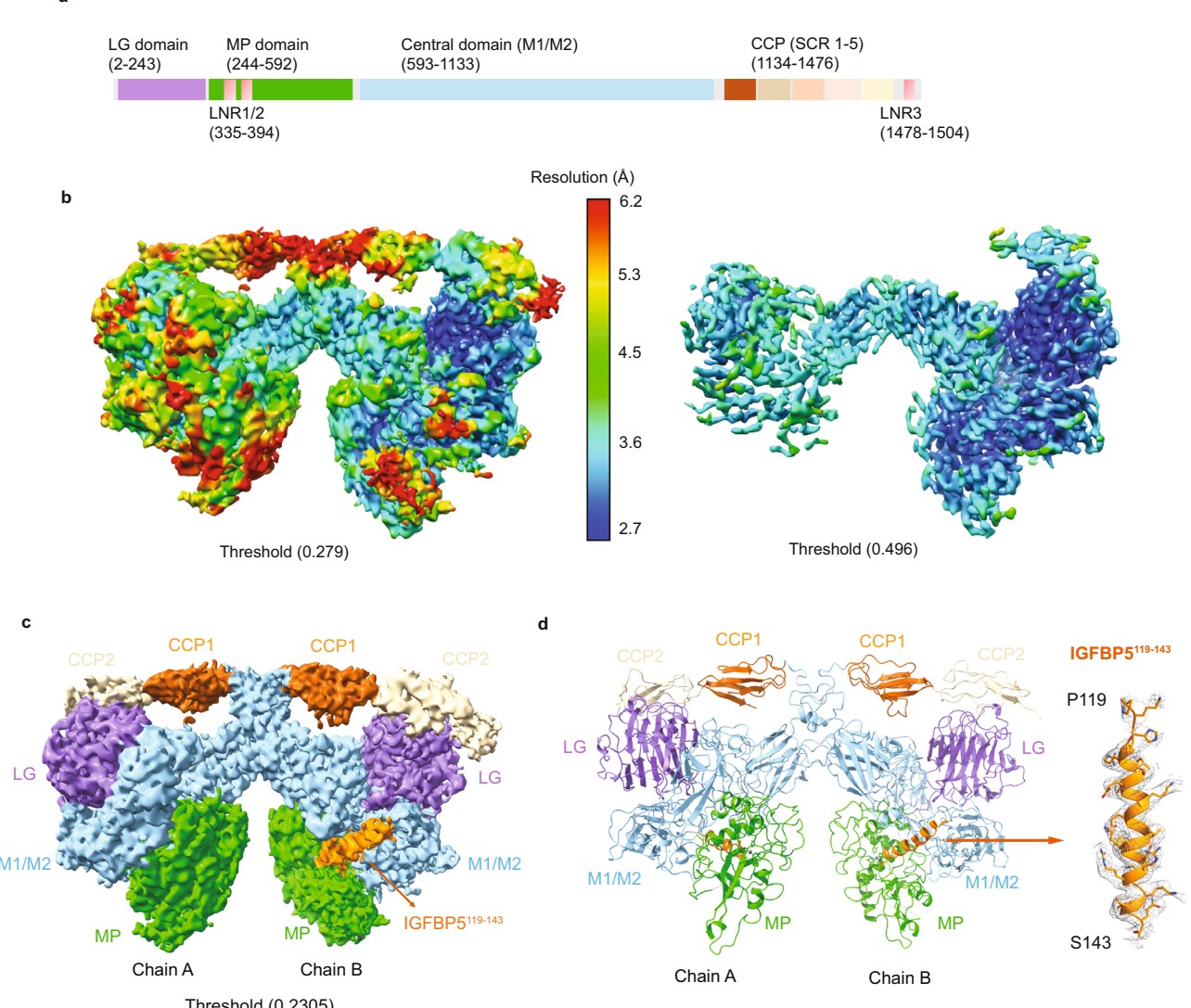

**Fig. 1 | AlphaFold2 facilitated structure determination of PAPP-A_BP5.**
**a** Schematic diagram of PAPP-A with domains highlighted by different colors (Laminin G-like domain (LG) purple, metalloprotease domain (MP) green, central domain (M1/M2) light blue, the Complement Control Protein domains or SCR (short consensus repeats) (CCP1) chocolate, (CCP2) wheat, (CCP3) peach puff, (CCP4) misty rose, (CCP5) khaki and the Lin12-Notch Repeats (LNR 1, 2, 3) salmon). **b** Cryo-EM density map of the PAPP-A_BP5 complex at an overall resolution of 3.28 Å, with two threshold levels used to highlight local resolution (left: 0.279 and right: 0.496). The structure showed that one monomer exhibits a higher resolution than the other and the PAPP-A core-domains have higher resolution than the C-terminal domains. The 3D cryo-EM map is colored by resolution (bar in the middle). The

local resolution is calculated by cryoSPARC v3. **c** PAPP-A_BP5 complex cryo-EM density map (EMD-26475) colored by different PAPP-A domains as in **a**, with IGFBP5 peptide highlighted in dark orange. The Lin12-Notch Repeats (LNR 1, 2, 3) and the CCP3-5 domains are not observed in the structure. The threshold level is set at 0.2305. The cryo-EM map was generated by ChimeraX (Supplementary Refs. 1, 2). **d** (Left) PAPP-A_BP5 complex structure represented in ribbon (PDB 7UFG) with PAPP-A domains highlighted in the same colors as in **a**, and IGFBP5 anchor peptide highlighted in dark orange. (Right) Enlarged IGFBP5 anchor peptide, from residues Pro119 to Ser143. Both mesh density and residue side chains are highlighted. The structure figures were prepared with Pymol and ChimeraX.

and cancer (ovarian, renal, breast, lung, gastric and pleural mesothelioma)[11]. In contrast, PAPP-A-deficient mice (PAPP-A KO) show a ~30% increase in longevity and exhibit a phenotype of proportional dwarfs[32,33]. Inhibition of PAPP-A also shows delayed progression of age-related pathology in various tissues and age-related thymic atrophy[33,34]. Collectively these observations underscore the importance of PAPP-A in regulating IGF activity in vivo, however its structure and substrate recognition mechanism remain poorly understood.

Here we report structures of PAPP-A in its substrate-unbound form and also in complex with substrate IGFBP5 (PAPP-A_BP5) determined using an AlphaFold[35] model as a starting template for building into our single-particle cryo-electron microscopy (cryo-EM) maps. Our work highlights the utility of using artificial intelligence (AI) predicted

models to facilitate the rapid determination of challenging protein structures. The complex structure provides mechanistic insights into PAPP-A domain packing, *trans*-dimer formation and substrate binding and selectivity.

## Results
### Structure of the PAPP-A_BP5 complex
Wild type (WT) PAPP-A (auto-cleaved)[16,36,37], the catalytically inactive enzyme PAPP-A (E483A) and WT PAPP-A2 were purified from HEK293 cells (Supplementary Fig. 2a–e). Consistent with previous reports[9,38,39], WT PAPP-A and PAPP-A (E483A) molecular weights correspond to highly glycosylated dimers whereas the WT PAPP-A2 paralogue is monomeric (Supplementary Fig. 2b–e). Gel-based proteolytic cleavage activity assay

coupled with liquid chromatography–mass spectrometry (LC-MS) confirmed that WT PAPP-A, but not PAPP-A (E483A), cleaves its substrates IGFBP4 and IGFBP5 specifically between residues M135/K136 and S143/K144 respectively (Supplementary Fig. 3a–c, e, f). Our study also demonstrated that WT PAPP-A2 cleaves IGFBP5 between residues S143/K144 (Supplementary Fig. 3d), in agreement with previous reports[9,36]. These observations confirmed the integrity and activity of the purified WT PAPP-A and WT PAPP-A2 samples.

To avoid autocleavage or cleavage of substrates, we used the catalytically inactive full-length (FL) PAPP-A (E483A) and FL IGFBP5 in the cryo-EM study (Supplementary Fig. 2a, f). The cryo-EM density map for the PAPP-A (E483A)/IGFBP5 complex was determined to an overall resolution of 3.28 Å and captured a dimer configuration (Fig. 1b, Supplementary Fig. 4a–e, Supplementary Table 1). The protein complex exhibits flexibility which results in limited resolution of local regions, with some C-terminal domains not being defined (Fig. 1b). The cryo-EM map shows good density for one monomer with weaker density for the other (Fig. 1b). Further analysis with multi-body refinement indicated the monomers are moving relative to one another, which explains the complex flexibility and dynamics (Supplementary Fig. 5 and Supplementary Movie 1). The majority of the PAPP-A structure was previously unknown, with ulilysin as the only reported PAPP-A homolog structure that shares conservation with the metalloprotease domain[40]. All the above made PAPP-A a challenging structure to solve de novo. We therefore utilized the predicted high confidence regions from the AlphaFold[35] model as a starting template which significantly reduced the time needed for model building (Supplementary Fig. 6a–d). The cryo-EM map combined with the AlphaFold model enabled tracing of the PAPP-A backbone and most side chains in the N-terminal core regions and the CCP1/2 domains (Supplementary Fig. 7). Extra densities at various reported glycosylation sites were observed but the densities were poorly defined, which prevented fitting of the sugars. No density was observed for LNR repeats 1 and 2 (residues 335 to 394) or for residues 1265 to 1547 which include domains CCP3-5 and the C-terminal LNR3 domain, indicating high flexibility for these regions. For IGFBP5 only a helical peptide encompassing residues from 119 to 143 was observed in the structure (see below) while the remaining IGFBP5 protein is disordered (Fig. 1c, d), so the complex cryo-EM structure will be named as PAPP-A$_{BP5}$ hereafter. The experimentally determined structure of the N-terminal core domains and the CCP1/2 domains of PAPP-A$_{BP5}$ largely agree with AlphaFold model, demonstrating the high quality of the predicted structure (Supplementary Fig. 8a–d).

Overall, the PAPP-A$_{BP5}$ dimer is a butterfly shaped molecule with the Laminin-G like domain (LG), metalloprotease domain (MP), and the central M1/M2 domain from the PAPP-A N-terminal core packed tightly to form the wings (Fig. 1c, d). No direct interactions are observed between the contiguous LG and MP domains. Instead, our structure shows the LG and MP domains are arranged on opposite sides of the central M1/M2 domain which serves as a scaffold for inter-domain interactions.

### The PAPP-A *trans* homodimer

The PAPP-A$_{BP5}$ structure is a swapped dimer in which the C-terminal regions of the M1/M2 domains crossover in space, leading to *trans*-dimerization (Fig. 1c, d, Fig. 2a–c). *Trans*-dimerization appears to be mediated by two loops: 1068-1078 (loop 1) and 1100-1111 (loop 2) from the M1/M2 domain (Fig. 2c). To test this observation, we purified chimeric proteins that replaced the above PAPP-A segments with monomeric PAPP-A2 sequences and monitored the state of dimerization (Supplementary Fig. 9a). Replacement of loop 1 with the corresponding amino acids from PAPP-A2 abolishes protein expression suggesting loop 1 is essential for proper folding and assembly of PAPP-A. The chimeric PAPP-A$^{1100-1111*}$ with loop 2

replaced by the corresponding amino acids 1125-1136 from PAPP-A2 expressed well and is in dynamic equilibrium between monomer and dimer, in contrast to WT PAPP-A which exists as a stable dimer (Fig. 2d). Previous biochemical reports suggested that PAPP-A is both a covalent dimer via disulfides and a non-covalent dimer[31,38]. There are 4 cysteines after loop 2: C1112 makes an intra-disulfide bond to C1125 in the extended dimer interface, C1130 makes an intermolecular disulfide bond to the neighboring molecule C1130, and C1135 makes an intra-disulfide bond to C1189 in the beginning of the CCP1. We therefore made a larger replacement in that region, by replacing PAPP-A$^{1100-1135}$ with PAPP-A2$^{1125-1162}$ (also called PAPP-A$^{1100-1135*}$) which resulted in a shift to a predominantly monomeric state (Fig. 2d). A four-cysteine mutation protein with C1112S, C1125S, C1130S and C1135S (PAPP-A$^{4C4S}$) yielded a mixture of monomers and dimers (Fig. 2d). In contrast the point mutation C1130S marginally affected the oligomerization state in agreement with previous reports (Supplementary Fig. 9b)[38].

The M1/M2 mediated *trans*-dimerization leads to distal interactions between the LG domain from one monomer and the CCP2 domain from the adjacent monomer (Fig. 2b). Key residues from the CCP2 domain (H1211, L1254 and F1257) insert into hydrophobic cavities in the LG domain (Fig. 2b). Curiously, PAPP-A variants with mutations L1254A/F1257A/H1211A or lacking the CCP repeats entirely (PAPP-A (1132)) are still able to homodimerize (Supplementary Fig. 9a, c), suggesting these interactions are secondary in *trans*-dimerization. Taken together, our studies suggest a flexible *trans*-dimer architecture, with dimerization primarily mediated by the M1/M2 domains and further stabilized via disulfide bonds and the interaction between the LG and CCP2 domains.

### PAPP-A substrate recognition of IGFBP5

To facilitate the understanding of IGFBP5 recognition, we further obtained the cryo-EM map of substrate-unbound PAPP-A (E483A) at ~3.35 Å (Supplementary Fig. 4f–j). This substrate-unbound map exhibits dimer configuration, but with a much lower resolution for the second monomer (chain A) compared with PAPP-A$_{BP5}$ (Supplementary Fig. 10a), and multi-body refinement suggested this should be due to the even larger movement between the two monomers (Supplementary Fig. 10b, c, Supplementary Movie 2). Nevertheless, we were able to reconstitute the complete structure for one monomer and a partial structure for the second monomer (Supplementary Fig. 10d, Supplementary Table 1). Same as the PAPP-A$_{BP5}$, the substrate-unbound PAPP-A structure shows the *trans*-dimer configuration (Supplementary Fig. 10e).

The PAPP-A$_{BP5}$ cryo-EM map contains density for a 6-turn helix that is neither found in the globally aligned AlphaFold prediction nor in the substrate-unbound PAPP-A (E483A) map (Supplementary Fig. 11a–d). This helical density extends from the metalloprotease domain active site out to the central domain (Fig. 1c, d, Fig. 3a, Supplementary Fig. 11f). IGFBP5 residues P119 to S143 were successfully fit into this density (Fig. 1d, Fig. 3a, Supplementary Fig. 11d). S143 from the anchor peptide extends into the metalloprotease active site and provides one of the four contacts to the Zn$^{2+}$ ion within the active site (Fig. 3a, Supplementary Fig. 11f). Intriguingly, the AlphaFold predicted IGFBP5 structure (AF-P24593)[35] also shows an extended α-helix (residues P119-S143, numbering without signal sequence) for the anchor peptide and the predicted structure aligns well with the helix observed in our PAPP-A$_{BP5}$ complex (Supplementary Fig. 11e). Further, the IGFBP5 regions flanking the anchor peptide are predicted to be flexible and this is consistent with the lack of density observed for them in the cryo-EM map for PAPP-A$_{BP5}$ (Supplementary Fig. 11e). To examine whether substrate binding introduces conformational changes to PAPP-A, we overlaid the structure of the substrate-unbound PAPP-A with the structure of PAPP-A$_{BP5}$. The two structures share the same overall domain architecture and no

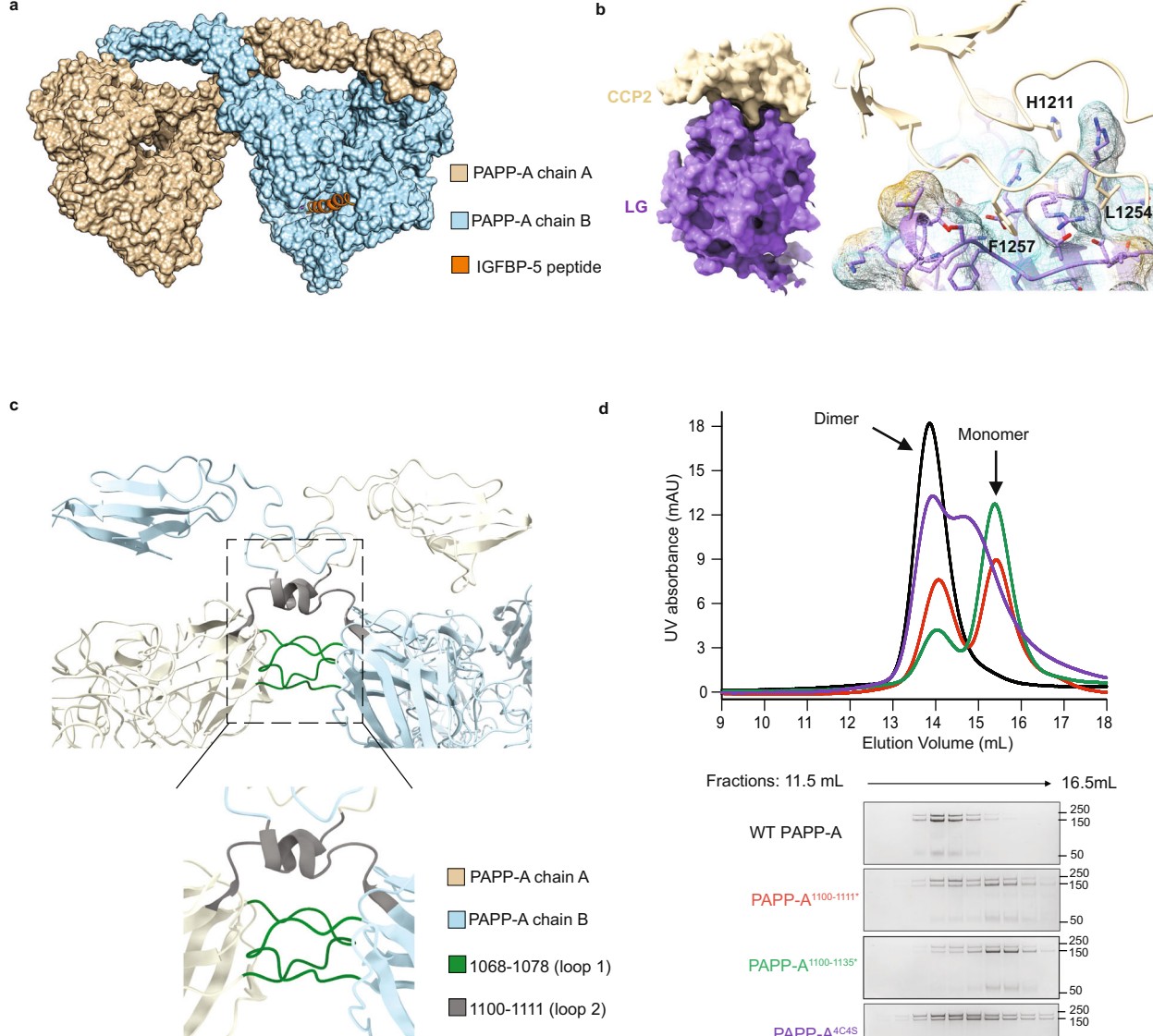

**Fig. 2 | PAPP-A dimerization mechanism. a** Overall cryo-EM structure of PAPP-A$_{BP5}$ in surface representation showed that PAPP-A exists as a dimer in *trans*. PAPP-A chain A and chain B are colored in wheat and blue, respectively. IGFBP5 anchor peptide is shown as a cartoon in dark orange. **b** (Left) Surface representation of the CCP2 and LG domain interaction. The CCP2 and LG domains are colored in wheat and purple, respectively. (Right) Close-up view of the interface between the CCP2 and LG domains. The crucial hydrophobic residues in the interface are highlighted as sticks. **c** (Top) Close-up view of the overall PAPP-A dimer interface with the two key loop regions highlighted in box. (Bottom) The enlarged interface with the two loop regions highlighted by different colors: residues 1068-1078 (loop 1) in green, and residues 1100-1111 (loop 2) in gray. **d** Size-exclusion chromatography (SEC) of recombinant purified WT PAPP-A, and the three monomer variants at 1.4 µM concentration. WT PAPP-A behaves as a stable dimer, whereas all three variants show shifts to monomer fractions. WT PAPP-A and monomeric variants were purified as auto-cleaved products. UV absorbance chromatograms (top) as well as Coomassie-stained fractions of reduced SDS-PAGE gel (bottom) are shown. Data points are representative of *n* = 3 independent replicates. Source data are provided as a source data file.

obvious conformational change was observed in the IGFBP5 binding groove (Supplementary Fig. 12a, b). There were slight differences in some loops around the substrate binding groove, but this is likely due to the limited resolution and the flexibility of these loops rather than representing specific conformational differences (Supplementary Fig. 12b). While we used the catalytically inactive PAPP-A (E483A) for structure work, the Zn$^{2+}$ coordination remains intact, which is a validation of the state of the active site (Fig. 3a, Supplementary Fig. 11f). However, there may be subtle differences in substrate binding due to this mutation that we are not able to distinguish. As aforementioned, multibody refinement of PAPP-A$_{BP5}$ and substrate-unbound PAPP-A showed that in the later structure, the two monomers have more obvious rotation (Supplementary Movie 1, 2),

suggesting the substrate association helps to stabilize the PAPP-A *trans*-dimer.

To confirm the ability of the IGFBP5 anchor peptide to bind PAPP-A, we synthesized fluorophore (Alexa Fluor 568) labeled and unlabeled IGFBP5$^{121-143}$ peptides to measure competitive binding using a florescence polarization (FP) assay. The labeled peptide binds WT PAPP-A at ~380 nM K$_D$ and is outcompeted by the non-labeled peptide, indicating on-target binding (Fig. 3b, Supplementary Fig. 13a). The Fam-labeled FL IGFBP5 binds PAPP-A (E483A) at a slightly higher K$_D$ (~250 nM measured by FP assay) (Fig. 3c), suggesting the anchor peptide is the primary binding site. The anchor peptide makes extensive interactions with both the metalloprotease and central M1/M2 domains (Fig. 3a, Supplementary Fig. 13b). The overall good surface complementarity

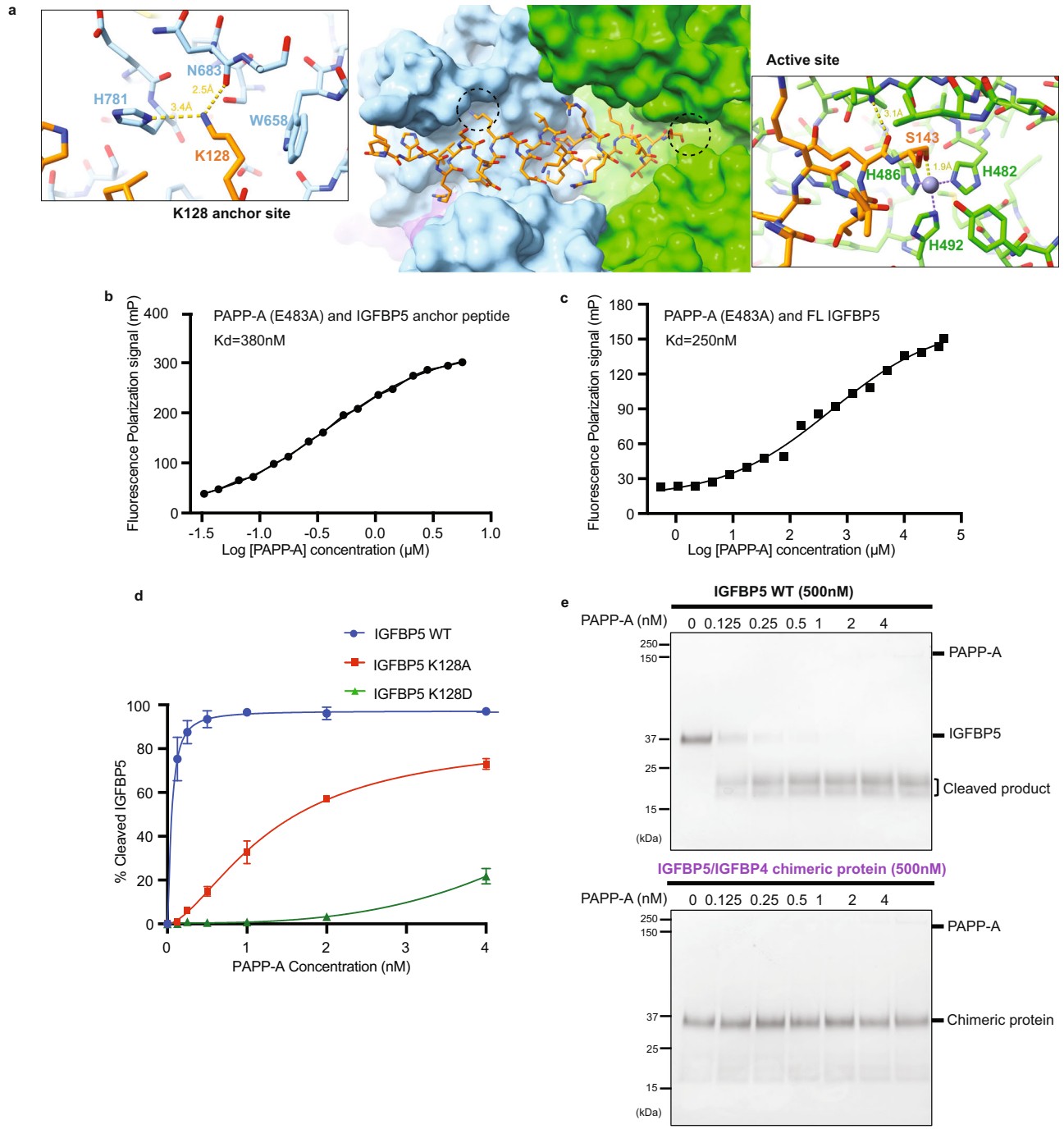

**Fig. 3 | Interaction mechanism between PAPP-A and IGFBP5. a** Structural depiction of the IGFBP5 anchor peptide binding to PAPP-A with the highlight of the K128 residue anchor site (left) and the active site (right). The structure shows that the IGFBP5 anchor peptide resides next to the zinc-coordination site of PAPP-A. **b** Binding affinity between the IGFBP5 anchor peptide and WT PAPP-A measured by fluorescence polarization (FP) assay. The N-terminal Alexa Fluor 568 conjugated IGFBP5[121-143] (5 nM) was incubated with serial dilutions of WT PAPP-A (5.65 μM to 33 nM) in PBS buffer at room temperature. **c** Binding affinity between the FL IGFBP5 and PAPP-A (E483A) measured by fluorescence polarization (FP) assay. FAM-labeled IGFBP5 (5 nM) was incubated with serial dilutions of PAPP-A (E483A) (10 μM−50 nM) in PBS buffer at room temperature. In **b**, **c** the FP values were recorded at 10 min. Data shown is the representative of three replicates. The data were analyzed in GraphPad Prism 9.1.2. **d** Quantification of the WT PAPP-A

proteolytic activity efficiency towards WT IGFBP5, IGFBP5 K128A and IGFBP5 K128D in the gel-based assay. Quantitative comparison was performed across samples from the same experiment with gels run in parallel. Bars are the mean ± standard deviation of $n = 3$ independent replicates (analyzed in GraphPad Prism 9.1.2). The single data representative of IGFBP5 K128A, IGFBP5 K128D were shown in Supplementary Fig. 13c. **e** SDS-PAGE gel-based WT PAPP-A proteolytic activity assay demonstrating the pivotal role of the IGFBP5 anchor peptide. WT PAPP-A has efficient cleavage towards WT IGFBP5, while the chimeric IGFBP5 with residues 121-143 replaced by the corresponding IGFBP4 residues 114-134 shows almost no response to WT PAPP-A. Data shown is the representative of 3 independent replicates. In **d**, **e**, 500 nM WT IGFBP5 or chimeric protein were incubated with WT PAPP-A in dose-response. The reactions were performed at 37 °C for 4 h. Source data are provided as a Source Data file.

suggests most amino acids at the interface are important for binding and recognition. Previous studies suggested a pivotal role for IGFBP5 K128 in substrate recognition[36]. Consistent with this previous observation, we found IGFBP5 K128A and K128D exhibit attenuated proteolysis by PAPP-A (EC$_{50}$ ~1.3 nM and ~15.3 nM respectively) relative to WT IGFBP5 (EC$_{50}$ ~53 pM) (Fig. 3d, Supplementary Fig. 13c). Deducing from our structure, the importance of IGFBP5 K128 is likely due to its ability to engage in hydrogen bonds (with the backbone carbonyl of PAPP-A N683 and with PAPP-A H781), as well as in hydrophobic interactions with PAPP-A W658 (Fig. 3a, Supplementary Fig. 13b). This anchoring interaction of the substrate in the central domain structurally explains the finding of how a lysine residing 16 residues from the scissile site is important for the binding of IGFBP5[36].

In terms of PAPP-A inhibition, proMBP is reported to form disulfide bonds with PAPP-A C381 and C652[22,30]. C652 resides in a surface exposed position on a loop near the IGFBP5 binding groove. C381 is in the LNR1/2 domain, which is not resolved in our structure, but based on the AlphaFold model this area is proximal to C652 and also to the substrate binding groove (Supplementary Fig. 14a). ProMBP therefore likely inhibits PAPP-A by sterically blocking access to the extended proteolytic binding groove.

## PAPP-A substrate selectivity

In addition to IGFBP5, PAPP-A also cleaves IGFBP2 and IGFBP4 in which IGFBP2 and IGFBP4 cleavage is IGF-dependent. The mechanism behind PAPP-A substrate selectivity however remains unclear. Sequence alignment of IGFBPs shows the IGFBP5 anchor peptide is not conserved in IGFBP2 or 4 (only IGFBP3 shows some similarity) (Supplementary Fig. 15), suggesting different substrate recognition mechanisms. To understand PAPP-A selectivity, we generated a chimeric IGFBP5/IGFBP4 construct with IGFBP5$^{121-143}$ replaced with a corresponding region from IGFBP4 (IGFBP4$^{114-134}$). The cleavage of this chimeric protein by PAPP-A was significantly attenuated when compared to WT IGFBP5 (Fig. 3e), demonstrating this anchor peptide mediates IGFBP5 recognition and subsequent cleavage by PAPP-A.

From our structure the LG and CCP2 domain trans-interactions appear to contribute to PAPP-A overall folding, although disrupting these interactions does not abolish dimer formation (Fig. 1, Fig. 2b, Supplementary Fig. 9c). We thus decided to examine its role in substrate selectivity and found that cleavage of IGFBP4 by the C-terminally truncated PAPP-A (1132) was significantly reduced compared to its cleavage by WT PAPP-A (EC$_{50}$ ~26 pM). The triple mutation L1254A/F1257A/H1211A also remarkably impairs PAPP-A's proteolytic activity for IGFBP4 (EC$_{50}$ ~2.6 nM). Intriguingly, the addition of domains CCP1 and CCP2 (PAPP-A (1267)) partially rescues PAPP-A's cleavage activity for IGFBP4 (EC$_{50}$ ~597 pM) (Fig. 4a, Supplementary Fig. 16a, c). Notably all variants retain similar cleavage activity for IGFBP5 (Fig. 4b, Supplementary Fig. 16b, d). Together, our data illustrate that the distal interactions between the LG and CCP2 domains are important for activity against IGFBP4 but not for IGFBP5. The observation that the presence of the CCP1/2 domains is important for IGFBP4 cleavage but not IGFBP5 is in line with the report that a natural PAPP-A variant, rs7020782 SNP (S1144Y), observed in the structure in a solvent-exposed location on domain CCP1 (Supplementary Fig. 14b) affects IGFBP4 cleavage but not IGFBP5[41].

The antiparallel dimer was reported to be important for IGFBP4 cleavage, but not for IGFBP5[38], eliciting the question of whether PAPP-A dimer formation is relevant to its physiological function. To address this, we analyzed the aforementioned PAPP-A monomer variants in the activity assay. Compared with WT PAPP-A (EC$_{50}$ ~ 26 pM), all the variants show impaired IGFBP4 cleavage activity (EC$_{50}$ of PAPP-A$^{1100-1111*}$ ~ 519 pM, PAPP-A$^{1100-1135*}$ ~954 pM, and PAPP-A$^{4C4S}$ ~402 pM) which correlates with the level of dimer formation (Fig. 2d, Fig. 4c, and Supplementary Fig. 17a). In comparison the variants retain the ability to cleave IGFBP5 with a similar level of activity as WT

PAPP-A (Fig. 4d, Supplementary Fig. 17b). Note that as a stable monomer PAPP-A2 can efficiently cleave IGFBP5 but not IGFBP4 (Supplementary Fig. 2d, e, Supplementary Fig. 3a, d)[9], which is concordant with our observation that IGFBP5 cleavage is dimer-independent. To test the selectivity arising from the PAPP-A trans-dimerization, inspired by prior analysis[38], we co-expressed a PAPP-A heterodimer consisting of FL PAPP-A (E483A) and the C-terminal truncated PAPP-A (1132) (Supplementary Fig. 18a). Intriguingly, compared with PAPP-A (1132) and PAPP-A (FL, E483A), the PAPP-A (FL, E483A)/PAPP-A (1132) heterodimer partially rescues the IGFBP4 cleavage efficiency (EC$_{50}$ ~ 466 pM) (Fig. 4e, Supplementary Fig. 18b), albeit the activity is still lower than WT FL PAPP-A. Importantly the PAPP-A (E483A)/PAPP-A (1132) heterodimer is as effective in cleaving IGFBP5 as PAPP-A (1132) and FL PAPP-A (Fig. 4f, Supplementary Fig. 18c). These data support the hypothesis that effective dimer formation, which is strengthened by the swapped trans-dimer, is important for IGFBP4 but not IGFBP5 cleavage.

A prior report by Weyer et al.[38] proposed an antiparallel PAPP-A dimer configuration in which the LNR3 domain of one molecule interacts with the LNR1/2 domains of the other molecule to form LNR centers, which are required for IGFBP4 cleavage. Our data also demonstrated that PAPP-A C-terminal regions including the LNR3 domain is essential for efficient IGFBP4 cleavage (Fig. 4a, Supplementary Fig. 16a), however, we could not confirm these interactions due to lack of density for these domains (LNR1-3) in our cryo-EM density map. We therefore used molecular dynamics (MD) simulations to predict the feasibility for the interaction between the LNR1/2 and LNR3 domains, by combining the PAPP-A$_{BP5}$ cryo-EM structure with the AlphaFold model to cover all domains (Supplementary Fig. 19a, Supplementary Data 1). The simulation result showed that the formation of the LNR centers by bringing LNR3 in close proximity to LNR1/2 is energetically unfavored in the PAPP-A$_{BP5}$ structure (Supplementary Fig. 19b, Supplementary Data 2). This in turn suggests that IGFBP5 cleavage by PAPP-A is independent of LNR center formation which may explain why those regions are disordered and therefore not observed in our PAPP-A$_{BP5}$ cryo-EM map. This reflects the finding of Boldt et al.[42] that the LNR domains function together to determine the substrate specificity of PAPP-A in that all LNRs are strictly required for IGFBP4 recognition but are not required for IGFBP5.

Overall, our structural and biochemical data shed light on three key determinants for PAPP-A substrate selectivity: (i) PAPP-A recognizes IGFBP5 through a unique IGFBP5 anchor peptide (~25 residue-long) which is not found in IGFBP4 or IGFBP2, and no significant conformational change of PAPP-A was observed upon IGFBP5 anchor peptide association, (ii) PAPP-A trans-dimerization is confirmed to be required for efficient IGFBP4 cleavage, but not IGFBP5, and (iii) the distal interaction between the LG and CCP2 domains supports proper PAPP-A folding which is important for IGFBP4 cleavage, but dispensable for IGFBP5.

## Discussion

We report structures of PAPP-A$_{BP5}$ and substrate-unbound PAPP-A, in which the structure determination was greatly facilitated by using the AlphaFold predicted model for PAPP-A[35]. We believe the use of AI protein structure prediction will be of significant benefit for elucidating the structure of many other novel challenging targets. The structure together with biochemical studies revealed PAPP-A domain functions that were previously unclear: (i) the central domain (M1/M2) has a critical role in supporting PAPP-A folding, mediates trans-dimerization and forms part of the substrate binding groove including a key anchoring interaction with IGFBP5 K128 which is important in IGFBP5 recognition, (ii) the interaction between the LG and CCP2 domains maintains proper PAPP-A architecture and is important for IGFBP4 cleavage.

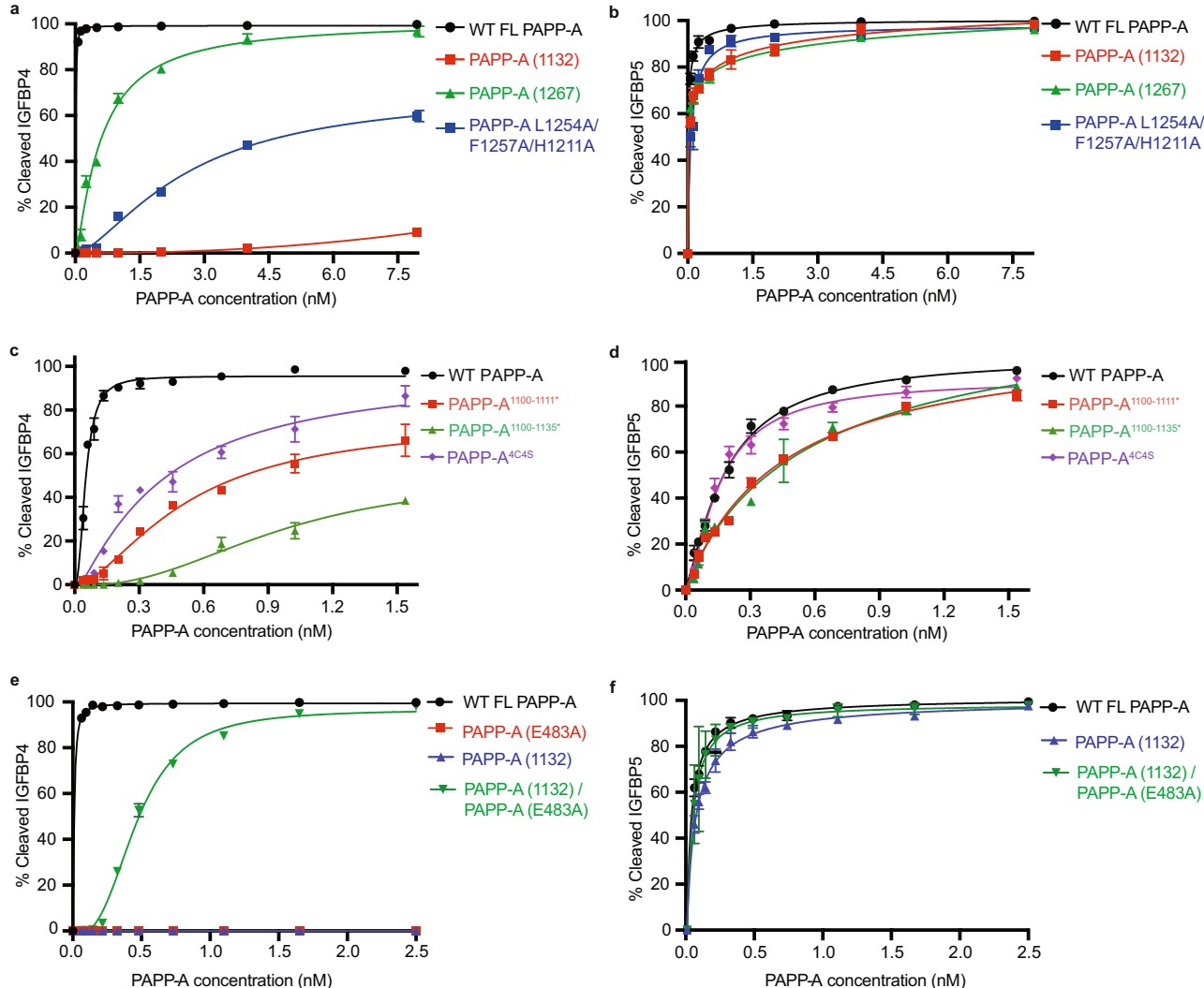

**Fig. 4 | PAPP-A substrate selectivity towards IGFBP4 and IGFBP5.**
**a**, **b** Proteolytic activity data suggest that the cleavage for IGFBP4, but not for IGFBP5, depends on the proper interaction between the LG and CCP2 domains. Quantifications of different PAPP-A truncations and the triple mutant (L1254A/F1257A/H1211A) proteolytic activity in gel-based assay towards IGFBP4 and IGFBP5 are shown in **a** and **b**, respectively. The single data representatives of the above gel-based assay are shown in Supplementary Fig. 16. **c**, **d** Proteolytic activity assay showed that PAPP-A dimer formation is crucial for IGFBP4 cleavage, but not IGFBP5. Quantifications of WT PAPP-A, PAPP-A[1100-1111*], PAPP-A[1100-1135*], PAPP-A[4C4S] proteolytic activity towards substrates IGFBP4 and IGFBP5 in the gel-based assay are shown in **c** and **d**, respectively. The single data representatives of the assay in **c**, **d** are shown in Supplementary Fig. 17a, b. **e**, **f** Co-expression of PAPP-A (1132) and

FL PAPP-A (E483A) significantly rescued IGFBP4 cleavage. Quantifications of FL PAPP-A, catalytic dead mutant PAPP-A (E483A), PAPP-A (1132), and the co-expressed PAPP-A (E483A)/PAPP-A (1132) heterodimer proteolytic activity towards IGFBP4 and IGFBP5 in the gel-based assay are shown in **e** and **f**, respectively. The single data representatives of the assay in **e**, **f** are shown in Supplementary Fig. 18b, c. For all proteolytic activity assay, 400 nM IGFBP4/700 nM IGF1, or 500 nM WT IGFBP5 were incubated with different proteases in dose-response. The reactions were performed at 37 °C for 4 h. In all quantifications, bars are the mean ± standard deviation of $n = 3$ independent replicates. Quantitative comparison was performed across samples from the same experiment with gels run in parallel. Source data are provided as a source data file.

Although the density in the dimer interface is not as strong as in the core domains, it is clear that PAPP-A exists as a *trans*-homodimer (Fig. 2a, Supplementary Fig. 10e). The AlphaFold model for monomeric PAPP-A appears more consistent with a *cis* conformation (Supplementary Fig. 6a, d). However, it is not our intention that the monomer prediction should be used to determine the dimerization mechanism of PAPP-A, especially when direct experimental evidence is available. Rather the contribution of AlphaFold lay in predicting the structures of the novel domains as well as the core domain packing, with sufficient accuracy to aid in density interpretation and model building. The cryo-EM multi-body refinement illustrated that PAPP-A is a flexible *trans*-dimer in the PAPP-A$_{BP5}$ complex (Supplementary Movie 1) and an even higher flexibility is observed in the substrate-unbound PAPP-A structure (Supplementary Movie 2), suggesting the substrate association

could contribute to the stabilization of the homodimer. In this regard further study of PAPP-A conformation dynamics could advance our understanding of PAPP-A dimerization. We also attempted to get the cryo-EM structure of WT PAPP-A, however the data quality was not sufficient to permit atomic resolution structure determination. As the $Zn^{2+}$ coordination in the active site in the structures we have obtained remains intact (Fig. 3a, Supplementary Fig. 11f), we infer the catalytic inactive mutant should not introduce any substantial confirmational changes in comparison to WT PAPP-A.

Our structural and biochemical data suggest a mechanism for PAPP-A substrate selectivity which further advances our understanding of how IGF signaling is tightly regulated by PAPP-A. Our findings show that the variants that interfere with dimerization or stability of the *trans*-dimer architecture affect the cleavage of IGFBP4

(Fig. 4a, c, and e). Based on this we hypothesize that besides binding to the N-terminal core domains, the IGFBP4/IGF complex also binds to the extended *trans* domains (including the CCP1 domain where residue 1144 locates) potentially across the dimer interface. In comparison, IGFBP5 primarily binds at the N-terminal core region of each PAPP-A monomer in the dimer. As our PAPP-A$_{BP5}$ cryo-EM map lacks density for the PAPP-A C-terminal domains and the majority of IGFBP5, and given the FP assay revealed that FL IGFBP5 has a relative higher binding affinity compared with the anchor peptide by itself (Fig. 3b, c), we cannot exclude the possibility of other interfaces between PAPP-A and IGFBP5. In this respect the IGFBP5 anchor peptide observed in the PAPP-A$_{BP5}$ structure is part of a larger flexible central linker domain (CLD) (Supplementary Fig. 11e). As for other IGFBPs the protease cleavage sites are located within the CLDs. A recent NMR study reported the CLD is naturally disordered in IGFBP2, and the binding of IGF1 further increases its flexibility, which potentially explains the IGF-dependent modulation of IGFBP2 cleavage on CLD[43]. In addition, a recent IGFBP3/IGF1/ALS ternary complex structure showed that the IGFBP3 binding with ALS is IGF-dependent, and the authors hypothesized that IGF1 association introduces a conformational change of IGFBP3 to enable the binding with ALS. However, in the structure of the IGFBP3, CLD was not resolved due to is flexibility[44]. These two reports agree with our observation in the PAPP-A$_{BP5}$ structure with respect to the extended CLD flexibility. In terms of IGFBP4 selectivity, the PAPP-A/IGFBP4/IGF complex structure would be highly desired to provide further insights on substrate selectivity and the mechanism for its IGF-dependency. In summary, as PAPP-A is reported to be associated with aging and multiple pathological diseases, the structural and biochemical data presented in this study provide new insights to enable drug discovery efforts targeting PAPP-A.

## Methods

### Plasmid construction

**PAPP-A and PAPP-A2.** The PAPP-A gene (Uniprot accession number Q13219) was codon optimized via the IDT Codon Optimization tool for expression in mammalian cells. The gene was divided and split into three regions (designated blocks 1, 2, and 3, respectively) and cloned into a pET-based in-house cloning vector with NotI/AscI restriction sites. A C-terminal GGSS-FLAG tag was added for affinity purification.

For mutagenesis, the appropriate gene block was used as a template and mutated with mutagenic primers and Platinum SuperFi II 2X Master Mix (Thermo Fisher: 12368010). The mutagenic PCR was then followed by a KLD reaction at 25 °C for 5 min using NEB KLD Reaction mix (NEB: M0554S) to ligate the linear mutated vector to a circular form and remove the methylated template vector. DNA was then transformed into DH5 *E.coli* (Zymo: T3007) and plated on LB agar plates supplemented with 100 μg/mL Carbenicillin. Colonies were picked into 5 mL LB + Carbenicillin (100 μg/mL), grown at 37 °C, and plasmid DNA purified with Zymo miniprep kit (Zymo: D4208T). Plasmid DNA was sequenced with CMV-F primer (5'-GGGCGGTAGGC GTGTACGGTGGGAG-3') and sequencing primer 2 (5'-CTGCATTCT AGTTGTGGTTTGTCC-3'). For final assembly, each gene cassette was amplified via PCR CMV-F and polyA-R primers. Block 1 was digested with NotI (NEB R3189L) and BsaI (NEB R3733L), block 2 with BsaI, and block 3 with BsaI and AscI (NEB R0558L) and all components cloned into an in-house expression plasmid digested with NotI/AscI + CIP (NEB M0525L). The DNA was transformed DH5 *E.coli*, colonies prepped and sequenced using gene-specific primers for validation. PAPP-A2 was cloned in a similar fashion with the gene being split into three block regions and cloned into a pET-based in-house cloning vector.

**IGFBP5.** Human FL IGFBP5 was codon optimized through IDT and ordered as a gBlock. We added an N-terminal secretion signal with a GAA linker, and a C-terminal 6X His- tag for IMAC purification. Mutagenesis was carried out in a similar way as the PAPP-A mutants. Clones

were sequence verified by Snapgene (version 3.2.1) and used in transient transfections.

The construct primers are all listed in Supplementary Table 2. Note that all the protein numbering after signal sequence.

### Expression and purification of PAPP-A, IGFBP5, and mutants

PAPP-A E483A was expressed as secreted protein in stably transfected Expi293F™ cells (Thermo Fisher, A14528) grown to a density of $2 \times 10^6$ cells/mL in Expi293F™ expression medium (Thermo Fisher, A1435101) at 37 °C. Cells were harvested by centrifugation at 2,000 g for 15 min (Sorvall LYNX 6000, Cat. no.75006590), and the supernatant was filtered (0.2 μm PES filter, Corning™ 430767) for purification. It was then purified on a Heparin HP column (5 mL bed volume; Cytiva 17-0407-01) in 1X PBS pH 7.4, 300 mM-NaCl to 1X PBS pH 7.4, 1 M NaCl salt-gradient. Fractions enriched for PAPP-A were identified by running the eluted fractions on the reduced SDS-PAGE, and further purified by size-exclusion chromatography on a Superose 6 Increase 10/300 column (Cytiva 29-0915-96) in 1X PBS pH 7.4. Purified protein fractions from size-exclusion chromatography were concentrated using 100 kDa MWCO concentrator columns (Millipore UFC901024) and stored at −80 °C for structural and biochemical analysis.

WT-PAPP-A, and PAPP-A variants used in the study were expressed as secreted proteins in the Expi293F cell system by transient transfection. Expi293F™ cells were grown to a density of $3 \times 10^6$ cells/mL in Expi293F™ media. Expi293F™ cells were transfected with the expression plasmid DNA at 1 μg/ml incubated with Expifectamine™ 293 transfection reagent (Thermo Fisher, A14525) at a ratio of 1:3.25 for 15 min at room temperature, and then grown at 37 °C in a humidified atmosphere with 8% $CO_2$ for 72 h in 1.6 L flasks at 150 rpm. For PAPP-A (1132)/PAPP-A (E483A) heterodimer formation, we co-expressed the two chains, with the plasmid of each chain mixed at 1:1 ratio to achieve a total concentration of 1 μg/ml, followed by incubation with Expifectamine™ reagent, and added the mixture to the Expi293F cells. Filtered conditioned media was purified by FLAG affinity chromatography (Genscript Anti-DYKDDDDK G1 Affinity Resin, Cat. No. L00432) and eluted with 0.8 mg/ml 3X-DYKDDDDK peptide (Genscript, Cat. No. RP21087) solubilized in 1X PBS. PAPP-A enriched fractions were further purified using a Superose 6 Increase 10/300 column in 1X PBS. Purified protein fractions from size-exclusion chromatography were concentrated and stored at −80 °C for proteolytic activity analysis.

WT-IGFBP5 and IGFBP5 mutants were expressed as secreted proteins through transiently transfected Expi293F cells at a density of $3 \times 10^6$ cells/mL in Expi293F media at 37 °C, 150 rpm in 1.6 L flasks. Following 48 h of transfection, cells were harvested by centrifugation at 2,000 g for 15 min, and the conditioned media was filtered (0.2 μm PES filter, Corning™ 430767) for purification. WT IGFBP5 and IGFBP5 mutants were purified with Ni-NTA agarose resin (Qiagen, Cat. #30210), washed with 15 mM imidazole pH 8.0, 0.5 M NaCl in 1X PBS buffer and eluted in 0.5 M imidazole pH 8.0, 0.5 M NaCl in 1X PBS buffer. The eluted fractions were concentrated and run on a Superdex 200 Increase 10/300 GL column (GE, 28-9909-44). The target fractions were concentrated using 10 kDa MWCO concentrator column (Millipore UFC901024) and stored at −80 °C until further use.

Recombinant human IGFBP4 (R&D Systems, Cat. no. 804-GB-025) was reconstituted in 1X PBS. Recombinant human IGF1 was sourced from Abcam (Ab270062) as lyophilized aliquots and reconstituted in 1X PBS, then aliquoted and stored at −80 °C.

All purifications were performed at 4 °C. Protein concentrations were measured on a Thermo Fisher NanoDrop (Cat. no. 13-400-519) based on their respective extinction coefficients and molecular weight values using absorbance at a wavelength of 280 nm.

### SEC-MALS

An Agilent 1200 Series Infinity II HPLC coupled to a DAWN Heleos II multi-angle light scattering detector and Optilab T-rEX refractive index

detector (Wyatt Technology) was used for size-exclusion chromatography multi-angle light scattering (SEC-MALS) analysis. PAPP-A, PAPP-A (E483A) samples (100 μL of 1 mg/mL), and PAPP-A2 samples (100 μL of 1 mg/mL, 3 mg/mL, and 6 mg/mL concentrations) were injected onto a Superdex200 Increase 10/300 GL column (Cytiva, Product: 28990944) at 0.5 mL/min for 60 min in PBS. Molecular weights were derived from analysis using Astra 7.0 software (Wyatt Technology) following calibration with BSA.

## Mass spectrometry−IGFBP4/5 cleavage
Cleavage reactions for IGFBP4 (500 nM) in the presence of IGF1 (500 nM), or IGFBP5 (500 nM) were initiated by the addition of PAPP-A (1 nM) in PBS at 37 °C. Cleavage of IGFBP5 by PAPP-A2 was performed under the same reaction condition. After 4 h, 5 μL of each reaction sample was injected into an Agilent 1200 Series Infinity II HPLC PLRP-S 300 Å on a 2.1 × 150 mm reverse phase column (Agilent, Part Number: PL1912-3501) equilibrated in water in 0.1% (v/v) formic acid and 5% (v/v) acetonitrile flowing at 0.6 mL/min with a column temperature of 80 °C. The protein peaks were resolved using a 5−50% gradient of acetonitrile in 0.1% (v/v) formic acid flowing at 0.6 mL/min and eluted into a Dual Agilent Jet Steam electrospray ionization source operating using Gas Temp at 325 °C, Drying Gas at 8 L/min, Nebulizer at 35 psig, Sheath Gas Temp at 400 °C, Sheath Gas Flow at 11 L/min, VCap voltage at 4500 V, and Nozzle Voltage at 1000 V. Protein ions were detected on an Agilent 6230 Time-of-Flight mass spectrometer operating in positive ion mode with a Fragmentor Voltage of 250 V and Skimmer Voltage of 65 V. Spectra were analyzed using MassHunter B.07 software and intact protein masses were derived by maximum entropy-based deconvolution algorithm.

## Structure prediction
The AlphaFold monomer prediction for PAPP-A was generated using the same trained models and inference procedure employed in CASP14[35]. This is described in Jumper, J. et al.[35]. Mean pLDDT (predicted local distance difference test) over the structure was used for ranking five models, and the model with the highest mean pLDDT was used throughout this study. The model confidence images in Supplementary Fig. 6b, c are taken from AF-Q13219.

The AlphaFold prediction for IGFBP5 was obtained from AlphaFold Protein Structure Database (Supplementary Refs. 5, 6). Accession AF-P24593.

## Cryo-EM
Prior to grid preparation, PAPP-A (E483A) and IGFBP5 were incubated on ice for 1 h at a molar ratio of 1:3. The complex was diluted with Bis-tris propane buffer pH 9.2 (Hampton Research HR2-103) to a final concentration of 0.25 mg/mL. A 3 μL drop of the sample was applied to a 1.2/1.3 C-Flat grid (Protochips, Product Number: CF-1.2/1.3-3CU50) that had been glow-discharged at 10 mA for 45 s in a PELCO easiGlow glow discharge cleaning system (PELCO, Product Number: 91000). Grids were plunge frozen in liquid ethane using the following settings on a Vitrobot Mark IV (Thermo Fisher Scientific): blot time 7.0-8.5 s, blot force 5, 10 °C, 100% humidity. The grids were imaged using an FEI Titan Krios (Hillsboro, Oregon) transmission electron microscope operated at 300 kV and equipped with a Gatan K3 Summit direct detector placed at the end of a BioQuantum energy filter (Gatan, Inc., model 1967), operating with a slit width of 20 eV. Automated data collection was performed with SerialEM software (Supplementary Ref. 7) at a nominal magnification of 105,000x, corresponding to a pixel size of 0.83 Å. A total of 16,453 movies were recorded using a nominal defocus range of −1.0 to −3.5 μm. Exposures were divided into 28-30 frames with an exposure rate of 23.8-25.2 e⁻/pixel/s and total exposure of 48-50 e⁻/Å². A total of 9,080,999 particles were selected for 2D classification in cryoSPARC v3. During the initial round of 3D classification, only one of the models appeared to have the correct size

and configuration corresponding to the PAPP-A stabilized homodimer, whereas the other classes were too small or flexible. A second round of 3D classification separated into three classes, with two classes resolving to a higher resolution and one class resolving to a lower resolution. The two higher resolution classes were merged and run through a second round of 2D classification for cleanup, resulting in a particle stack of 245,018. This was followed by non-uniform refinement with C1 symmetry imposed and global contrast transfer function (CTF) optimization enabled, resulting in a final map with a resolution of 3.28 Å using the gold-standard FSC = 0.143 criteria. Data processing statistics are provided in Supplementary Table 1.

Grid preparation and imaging protocol of substrate-unbound PAPP-A (E483A) was the same as PAPP-A(E483A)/IGFBP5. A total of 9,982 movies were recorded using a nominal defocus range of −1.0 to −3.5 μm. For substrate-unbound PAPP-A (E483A) data processing, movie frames were patch-motion-corrected and dose-weighted using cryoSPARC v3. CTF parameters were estimated from the dose-weighted aligned movie frames with Patch CTF. A total of 2,145,158 particles were selected using a blob template in cryoSPARC v3. The particles were subjected to 2D classification resulting in 14,229 particles. Ab initio models were generated from this particle stack: one clear dimer and three junk or monomeric classes. A round of heterogeneous classification using the dimer and junk classes, followed by homogeneous refinement was used to generate a 3D volume from which 2D templates were created for template matching. A total of 4,032,492 particles were selected and subjected several rounds of 2D and heterogenous classification resulting in 338,320 particles exhibiting the characteristic stabilized dimer conformation. The final particle stack was used for non-uniform refinement with global CTF refinement resulting in a final map with a resolution of 3.35 Å using the gold-standard FSC = 0.143 criteria. Data processing statistics are provided in Supplementary Table 1.

Multi-body analysis was used to evaluate dimer flexibility. Particles from cryoSPARC were exported to RELION v3 using pyEM (Supplementary Ref. 8). A consensus refinement was generated using the PAPP-A$_{BP5}$ or substrate-unbound PAPP-A (E483A) cryoSPARC volumes as a starting model. Two soft-masks were generated for each half of the map from the consensus model and used as an input for multi-body analysis with local angular and translational search restricted to 30 degrees and 6 pixels respectively. Movies were generated with relion_flex_analyze for the first two eigenvectors.

For the de novo AlphaFold FL WT-PAPP-A predicted structure, the output file was monomeric with a compact core containing the LG domain, MP domain, the central region (M1/M2) then a bent back extended stretch of CCP domains culminating in a C-terminal LNR domain. The Cryo-EM map for the PAPP-A (E483A)$_{BP5}$ complex however only showed density for the core domains (LG, MP, M1/M2) and for the first two CCP domains. As the core domains were better defined and as we also observed a crossed dimer in the map, a truncated AlphaFold model including the core domains and excluding the CCPs and C-terminal LNR domain was initially docked using Phenix Dock in Map and further fit and refined using the Phenix programs CryoFit and Real-space Refinement. The lower resolution CCP domains were then fit and refined using manual adjustment in COOT followed by refinement using Real-space Refinement. The same procedure was used to solve the substrate-unbound PAPP-A (E483A) structure. During the model building and refinement phase of the PAPP-A (E483A)/IGFBP5 data set, a helical density was observed in the protease binding groove. The IGFBP5 peptide 119-143 was fit and refined to this density. Analysis and model validation for both structures were performed using COOT and the Phenix validation tool (Supplementary Ref. 9, 10). Structure analysis is performed with ChimeraX and Pymol. Model building statistics are provided in Supplementary Table 1.

In terms of the protein numbering, we chose to number both PAPP-A and IGFBP5 after the signal sequence, which is a different

numbering to that used in the UniProt database (UniProt Q13219 for PAPP-A, and UniProt P24593 for IGFBP5). The first 80 amino acids of PAPP-A and the first 20 amino acids of IGFBP5 are the signal peptides that are naturally cleavage off during protein production. We number the residues of both proteins after the signal sequence to be consistent with previous reports[11,22,24,28,31,36,38,42].

## Labeling of full-length recombinant IGFBP5

Recombinant IGFBP5 was labeled with FAM-maleimide, 6-isomer (Lumiprobe, Cat#24180) following the manufacturer's recommended protocol. IGFBP5 was reconstituted to a concentration of 3 mg/mL using 1X PBS, pH 7.4. Tris-carboxyethylphosphine (TCEP) dissolved in molecular biology grade water at a stock concentration of 1 mM was added to the IGFBP5 solution to a final concentration of 0.1 mM The sample was kept at room temperature for 20 min to reduce disulfide bonds. FAM-maleimide, 6-isomer dissolved in DMSO at 1 mg/mL was added to the sample and allowed to incubate at 4 °C, overnight. Excess dye and reducing agent was then removed by gel filtration using a Superdex 200 Increase 10/300 GL column (GE, 28-9909-44) equilibrated in 1X PBS, pH 7.4.

## Fluorescence polarization assay

Binding of IGFBP5 anchor peptide to PAPP-A was measured using fluorescence polarization on a CLARIOstar plate reader (BMG Labtech, Cat. No. 0430-101) using 384-well fluorescence assay plates (Corning, Cat. No. 4514). Measurements were made using an optical path consisting of an 540-20 nm excitation filter, LP 566 dichroic mirror, and an 590-20 nm emission filter. The IGFBP5 anchor peptide synthesized with an N-terminal labeled Alexa Fluor 568 dye (Wuxi AppTec) was used at a final concentration of 5 nM in PBS for gain and focal height adjustments to have a target minimal polarization value of 10 mP. For direct binding measurement, PAPP-A was serially titrated onto the assay plate in PBS. Binding was initiated with the addition of labeled IGFBP5 anchor peptide to achieve a final concentration of 5 nM labeled peptide in 20 µL total volume per well before measuring fluorescence polarization values on the plate reader using 200 flashes per read. For competitive binding experiments, unlabeled IGFBP5 anchor peptide (Wuxi AppTec) was serially titrated onto the assay plate in PBS followed by the addition of labeled IGFBP5 anchor peptide. Binding was initiated with the addition of PAPP-A to achieve a final concentration of 100 nM PAPP-A and 5 nM labeled IGFBP5 anchor peptide in 20 µL total volume per well before measuring fluorescence polarization values on the plate reader using 200 flashes per read.

PAPP-A binding to FAM-labeled IGFBP5 was measured by fluorescence polarization on a CLARIOstar plate reader (BMG Labtech, Cat# 0430-101) using 384-well fluorescence assay plates (Corning, Product Number: 4514). Measurements were made using an optical path consisting of a 482-16 nm excitation filter, LP 504 dichroic mirror, and a 530-40 nm emission filter. FAM-labeled IGFBP5 at a final concentration of 5 nM in PBS was used for gain, focal height, and baseline adjustments. For direct binding measurements, PAPP-A was serially titrated from 10uM to 0.5 nM onto the assay plate in 1X PBS. The time point was initiated with the addition of FAM-labeled IGFBP5 to a final concentration of 5 nM per well. Polarization values were taken every 65 s on the plate reader at 220 flashes per read.

The equilibrium dissociation constant $K_D$, was determined following a ligand-receptor kinetics model that describes $K_D$ as the receptor concentration when half of all receptors are bound to ligand at equilibrium (Supplementary Ref. 11). In our fluorescence polarization system, that is the protein concentration when anisotropy is at half of the maximum. The polarization amplitude vs log [PAPP-A] curve taken at the 10-min time point was fitted to a non-linear regression dose response model using Prism (GraphPad Software, Prism version 9.1.2) and the $EC_{50}$ calculated was acknowledged to be the $K_D$.

## Size-exclusion chromatography assay

For the SEC assay to examine dimerization mechanism, purified WT PAPP-A, PAPP-A (E483A), PAPP-A2 WT, PAPP-A monomeric mutants, and C-terminal truncation constructs (PAPP-A (1132) and PAPP-A (1267)) were thawed and centrifuged at 15,000 g for 5 min at 4 °C to remove any potential precipitates. Concentration of the PAPP-A proteins were normalized to 1.4 µM, then 0.2 mL of each protein was injected onto a Superose 6 Increase 10/300 column (Cytiva 29-0915-96) connected to an AKTA Pure (GE Healthcare). The system was run at 0.5 mL/min for 1 h using 1X PBS as the mobile phase. UV280 measurements were obtained directly from the instrument. The fractions from retention volume between 11.5-16.5 ml were run on a 4-12% Bis-Tris SDS-PAGE, stained with Coomassie Protein Stain (InstantBlue® ab119211) and destained with Milli Q Water.

## Gel-based proteolytic activity assay

In vitro cleavage reactions were carried out in a total reaction volume of 30 µL in 1X PBS. IGFBP4/IGF1 was used at a ratio of 1:1.75 with a final concentration of 400 nM IGFBP4 and 700 nM IGF1, pre-incubated at room temperature for 25 min prior to the reaction with WT PAPP-A or PAPP-A mutants. IGFBP5 was used at a final concentration of 500 nM for proteolytic cleavage reactions. The concentration range of serial-dilutions of WT PAPP-A or PAPP-A mutants was decided based on the different proteolytic cleavage assays. Proteolytic reactions were performed at 37 °C for 4 h. All reactions were quenched by the addition of 5 mM EGTA (Fisher Scientific, AAJ60767AD).

For the catalytic dead mutant PAPP-A cleavage activity test, PAPP-A E483A or WT PAPP-A were added to 8 µM IGFBP4 or 8 µM IGFBP5 substrate in a dose dependent manner, at 37 °C for 4 h or on ice for 4 h. All reactions were quenched by the addition of 5 mM EGTA (Fisher Scientific, AAJ60767AD).

The quenched reactions were applied to 4-12% Bis-Tris SDS-PAGE gel using 1 X MES as running buffer. The Intact substrate and co-migrating cleavage products were separated on the reduced SDS-PAGE gel. Cleavage efficiency was determined by integrating band intensities of intact substrate bands (IGFBP4/IGFBP5) using Image Lab (Bio-Rad, Version 6.1.0) and calculating the percentage of cleavage against intact IGFBP4/IGFBP5 controls. Percentage of cleaved substrate was plotted against the concentration of protease to determine the $EC_{50}$ values. Average and standard deviation of the 3 independent replicates were calculated. Calculations were performed using Microsoft Excel (Version 16.59). The $EC_{50}$ values were determined by fitting the % cleavage vs PAPP-A concentration to a non-linear regression dose response model using Prism (GraphPad Software, Prism version 9.1.2).

## Molecular dynamics simulations

Before running MD simulations and subsequently free energy calculations, we added the missing regions in the homodimer complex. We used the predicted AlphaFold structure as a template to build in several regions in both PAPP-A chain A (residues:415-501 including LNR1/2, 685-688 in the M1 region, 765-774 in the M1 region, and 1347-1627 including CCP3, CCP4, CCP5, and LNR3 regions) and chain B (residues:434-492 including LNR1/2 region, 1254-1264 in the CCP1 region, and 1345-1672 including CCP3, CCP4, CCP5, and LNR3 regions) that were not detected in the cryo-EM structure. Note the MD simulations utilized protein sequence numbering which includes the 80 amino acid signal sequence.

To model in M1 and LNR1/2 regions, we superimposed our cryo-EM structure to the predicted AlphaFold using Needleman-Wunsch alignment algorithm (Supplementary Ref. 12) with BLOSUM-62 matrix, which are incorporated in Chimera. However, to maintain the *trans* conformation revealed in the cryo-EM, we superimposed the C-terminus (from 1214-1584) of AlphaFold to the ones in the cryo-EM structure to complete the homodimer. Subsequently, we modeled in missing sidechains using Dunbrack-2010 rotamer library

(Supplementary Ref. 13), which is incorporated in Chimera to prepare the complex for simulations (shown in Supplementary Fig. 19a). Finally, the refined protein complex was immersed in ~208 K water molecules in a simulation box of 190 Å × 190 Å × 190 Å. We then neutralize the system and added 0.15 M NaCl for simulations.

To refine our model construct, the system was minimized using 500 steps of energy minimization according to the steepest descents algorithm incorporated in GROMACS (Supplementary Ref. 14). The optimization was followed by an MD simulation in a canonical ensemble, where the system was heated gradually from 0 K to 310 K in 20 ps. Then, an MD simulation in an isobaric-isothermal ensemble was carried out for 80 ps with maintaining the pressure at 1 bar to relax the simulation box. During these whole pre-equilibration steps, the positional restraints were placed on all heavy atoms and Zn ions using 47.8 kcal.mol$^{-1}$Å$^2$, which were progressively reduced to 0 kcal.mol$^{-1}$Å$^2$ for the final equilibration step. Subsequently, two separate classical MD simulations in isobaric-isothermal ensemble for 120 ns and 120 ns, respectively were performed to equilibrate the protein construct.

To examine whether the interaction between LNR3 and LNR1/2 would play a key role in cleavage of IGFBP5, we performed a free energy calculation to estimate the affinity between two regions using umbrella sampling method (Supplementary Ref. 15). We inserted the bias forces on the distance between the LNR1/2 of chain B (the center of mass of Cαs for residues 335-394) and the LNR3 of chain A (the center of mass of Cαs for residues 1478-1504). To do this, we sampled 17 windows/conformations with 2 Å increment in distance between LNR1/2$_B$ and LNR3$_A$, varying from 19 Å to 53 Å. We performed a set of 5.2 ns MD simulation on each window to relax the conformations. A set of restraints with a constant force of 0.8 kcal.mol$^{-1}$Å$^2$ on the distance between LNR1/2$_B$ and LNR3$_A$ was used during these calculations. Another set of positional restraints with force constant of 23.9 kcal.mol$^{-1}$Å$^2$ was placed on the backbone atoms of residues that were resolved by cryo-EM to avoid undesirable deviation from the experimental structure. Thus, the residues built in the homodimer structure from the AlphaFold2 structure were relaxed during these free energy calculations. To obtain 17 different conformations, we steadily encouraged the LNR3$_A$ to go towards LNR1/2$_B$ by applying the force constant of 0.05 kcal.mol$^{-1}$Å$^2$ using Plumed-2.8 (Supplementary Ref. 16). The free energy profile (Supplementary Fig. 19b) was obtained by the weighted histogram analysis method (Supplementary Ref. 17). The statistical errors were evaluated by the bootstrap method (Supplementary Ref. 18) as shaded with pink in the Supplementary Fig. 19b.

In all simulations and free energy calculations, PAPP-A chains, the IGFBP5 peptides and ions were described using the Charmm36m (Supplementary Ref. 19) parameter set. Water was described using the TIP3P model. The temperature was maintained at 310 K using a velocity-rescale (Supplementary Ref. 20) thermostat with a damping constant of 1.0 ps for temperature coupling and the pressure was controlled at 1 bar using a Parrinello-Rahman barostat algorithm (Supplementary Ref. 21) with a 5.0 ps damping constant for the pressure coupling. Isotropic pressure coupling was used during these calculations. The Lennard-Jones cutoff radius was 12 Å, where the interaction was smoothly shifted to 0 after 10 Å. Periodic boundary conditions were applied to all three directions. The Particle Mesh Ewald algorithm (Supplementary Ref. 22) with a real cutoff radius of 10 Å and a grid spacing of 1.2 Å was used to calculate the long-range coulombic interactions. A compressibility of $4.5 \times 10^{-5}$ bar$^{-1}$ was used to relax the box volume. In all the above simulations, water OH bonds were constrained by the SETTLE algorithm (Supplementary Ref. 23). The remaining H-bonds were constrained using the P-LINCS algorithm (Supplementary Ref. 24). All simulations were carried out using GROMACS (Supplementary Ref. 25).

The initial and final configurations for MD-Simulation could be found in Supplementary Data 1 and 2, respectively.

## Statistics and reproducibility

Data are presented as mean values ± SD (standard deviation) or ±SEM (standard error of the mean), calculated using Microsoft Excel 2022/version 16.59 and GraphPad Prism 8 version 9.1.2. Derived statistics correspond to analysis of averaged values across independent replicates. For the % cleavage activity curves, non-linear regression dose-response model was used to determine the EC$_{50}$ values.

## Reporting summary

Further information on research design is available in the Nature Research Reporting Summary linked to this article.

## Data availability

The data that support this study are available from the corresponding authors upon reasonable request. The atomic coordinates for PAPP-A and the PAPP-A IGFBP5 complex have been deposited in the Protein Data Bank under the accession numbers 8D8O and 7UFG respectively. The Cryo-EM maps for each have also been deposited with the accession numbers EMD-27253, and EMD-26475, respectively. For MD-simulation, the initial and final PDB configurations are provided as Supplementary Data 1 and 2, respectively. The full data set is stored locally and could be provided upon request. Source data are provided with this paper.

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

## Acknowledgements

We thank our AbbVie colleague Yuliya Kutskova for support with the purification of recombinant PAPP-A proteins. We thank our Calico colleagues Reyna Simon, Yao Wong, Michelle Chan, Veronica Kuiper and Georgios Koukos for feedback on the manuscript. We thank Kyoungh-wan Lee for his extensive optimization of cryo-EM grid preparation and imaging conditions.

## Author contributions

These authors contributed equally: R.A.J., J.S., and K.T. These authors jointly supervised this work: D.E. and Q.H. D.E. and Q.H. conceptualized and supervised the work. K.T., A.H.N., J.X., R. J., V.S., A.H.S and J.J. provided additional supervision. C.O. and C.X. performed cryo-EM data collection and data processing, and R.J., R.A.J. performed the cryo-EM structure solution and R.A.J. performed model building. K.T. and J.J. performed AlphaFold structure prediction, A.M. performed MD-simulation analysis and J.S., J.CK.W., C.R., C.G., C.L.S., and Q.H. performed experiments. Q.H., D.E., R.A.J. and R.J. wrote the manuscript. All authors contributed to review and editing.

## Competing interests

J.S. J.C.K.W, J.X., A.M., A.H.N., C.R., C.G., D.E. and Q.H. are employees of Calico Life Sciences LLC and declare no other competing interests. R.A.J, R.J., C.L.S and V.S. are employees of AbbVie and own AbbVie stocks. K.T. and J.J. are employees of DeepMind. J.J. has filed non-provisional patent applications 16/701,070 and PCT/EP2020/084238, PCT/EP2021/072552, PCT/EP2021/082684, PCT/EP2021/082696, PCT/EP2021/082698 and PCT/EP2021/082707, each in the name of Deep-Mind Technologies Limited, each pending, relating to machine learning for predicting protein structures. C.O. and C.X. declare no competing interests. AbbVie, Calico, and DeepMind contributed financial support for the study and participated in the design, conduct, data analysis, review and approval of the publication.
