## [Peer Review File · Nature Communications]

Structure of the PAPP-ABP5 complex reveals mechanism of substrate recognitionREVIEWER COMMENTS

Reviewer #1 (Remarks to the Author):

RA Judge and colleagues report the structure of the metalloprotease PAPP-A protein in complex with its substrate IGFBP5 in attempt to delineate the mechanism of substrate recognition by PAPP-A. The study potentially addresses an interesting question regarding how PAPP-A selectively recognizes IGFBP5. However, the current set of structural and biochemical data run in a deficient way that may require additional clarification. Particularly, the lack of comparison with a substrate-free wild-type PAPP-A structure, and associated functional tests, leave it unclear whether the presented structure informs the exact mechanism of substrate selection or just provides one possible mechanism that is not sufficiently examined. It remains unclear if any conformational changes are required for or induced by the substrate recognition, which might be a crucial factor contributing to the substrate selectivity. This is worsened by the uncertainty between two possible models of cis against trans-dimer. Although this study looks like having some potential to go deeper and be conclusive, the authors appear to stop with a limited amount of data that provide less certainty for the underlying mechanism and the true structural model. Thus, I don't see that the current presentation meets the high quality commonly observed in a paper published in Nature Communications. Therefore, I would like to delay my recommendation until the authors answer the following enlisted questions, given that the authors might improve the manuscript to reach a more convincing conclusion and insights rather than leaving too many open questions around a very specific area.

(1) What is the structure of substrate-free wild-type PAPP-A? Is it still forming trans-dimer? How different is it as compared to the structure of IGFBP5-bound PAPP-A mutant? Any allosteric effects are involved in the substrate recognition? Any conformational changes are induced by substrate binding and contribute to substrate selectivity?

(2) How is the AlphaFold2-predicted PAPP-A compared to the IGFBP5-bound and substrate-free PAPP-A? How different are they? Any discussion on the implications of the difference might be considered? It would be nice to present the comparison in superimposed models.

(3) How much the E483A mutation effects on the conformation of PAPP-A? Does it keep the native structure of PAPP-A or induce certain structural rearrangement? It would be no help to take it as a given for a mutant structure and interpret it as if it is a wild-type structure. This creates more open questions than answers them.

(4) If E483A mutation does induce structural changes, as expected for many enzymes or proteins, then the authors need to discuss its implication in substrate recognition. It may be necessary to point out the caveat of interpreting such a mutant structure in terms of substrate selectivity.

(5) The domain swap architecture is interesting. Is this architecture also preserved in the wildtype PAPP-A dimer structure? How certain is the domain swap architecture, given the lower resolution in the neck region of the swap domains (shown in Fig. 1b and Fig. 2a)? Fig. 2a appears to contradict Extended Data

Fig. 6. No better explanation and solid evidence why the authors choose the domain-swapped model? The data presentation or discussion do not appear to be sufficiently convincing in the regard.

(6) The authors need to show the Zinc density and the side chains at the active site, as a direct validation of the state of active site. What about the validation using FSC to measure how good the atomic model fitting the cryo-EM map? How valid is the currently reported 3.3 Å overall resolution? The local-resolution map of the apo mutant PAPP-A should also be shown.

(7) How about the prediction of the dimer structure by AlphaFold-Multimer? Does it also predict a trans-swapped dimer model or still a cis-dimer? What about the wild-type PAPP-A dimer structure? How much is the cis/trans confusion affected by the E483A mutation?

(8) Most importantly, due to the modeling uncertainty and lack of a wildtype structure comparison, how much the reported complex represents a functional complex is a major open question that is central to the subject of this study? Without addressing this issue to a sufficient level, it may not be a good fit for a journal like Nature Communications, where readers are expecting more certain answers rather than just tentative models with some major uncertainty.

(9) Both the title and abstract did not mention that the structure is analyzed with mutant PAPP-A, with considerable uncertainty in the dimer architecture, which felt a little bit deceptive. The fact that AlphaFold2 did not help clarify this, but instead, make the uncertainty greater by predicting cis-dimer, leaves it somewhat ironic, in contrary to the rhetoric in the abstract that emphasizing “by leveraging the power of AlphaFold2”. If the authors cannot fully conclude on trans-dimer against cis-dimer, why make such a conclusion in the abstract and claiming that they have solved the structure of trans-dimer? I think the right way to handle this is to improve the local resolution of the neck region of the density map by either adding more data or by better 3D classification approach. Eventually, a high-resolution cryo-EM density at this critical local region itself presents the most interpretable data and evidence to solve the modeling ambiguity.

My last comment is that to solidify the mechanism of dynamic interactions between protein counterparts often requires determination of multiple conformations at representative functional states/stages. For PAPP-A and IGFBP5, this does not seem to be an exception. Limited by insufficient resolution on the critical local region and inconsistency between human interpretation and AlphaFold2 prediction, this study opens more questions than it answers. Most problematically, the final structural model is unfortunately uncertain, leaving the mechanistic interpretation too speculative and preliminary to publish in any serious research journal. The authors really need put on some efforts to at least remove the structural ambiguity before deciding on final publication.

Reviewer #2 (Remarks to the Author):

The manuscript by Judge and colleagues reports on structural studies by cryo-EM aimed at deciphering the mechanism employed by the metalloendopeptidase PAPP-A to selectively cleave its highly specific substrates, viz. three out of six insulin-like growth factor binding proteins (IGFBP-2, -4 and -5). Given that the enzyme does not cleave any other substrates, it has been long speculated which the molecular determinants underlying this very high specificity would be. Here, Judge and colleagues, assisted by AlphaFold-frontman John Jumper, come up shedding light on the solution with a partial structure of the enzyme in complex with a short helical fragment of substrate IGFBP-5, i.e. a product complex. This is a very interesting and long-awaited structure, which clearly should deserve publication in Nature Communications. However, there are a series of issues that I would need the authors to clarify/amend before definitive acceptance.

-The reported structure is a product complex of most of PAPP-A with a very small fragment of IGFBP-5 (25 residues), i.e. not the entire proteins. Please, make this clear in the title and throughout the text.

-Please, adopt the numbering of the corresponding UniProt entry (UP Q13219) for clarity. The current numbering lags 80 positions behind, which is misleading. The same holds for IGFBP-5. It is very useful to have a homogeneous numbering throughout the distinct databases.

-The cryo-EM map shows clear density for most of the N-linked glycosylations. For the sake of model completeness, I would invite the authors to model them, even if only for one NAG unit.

-The structure encompasses nine domains from the entire structure, all except the three C-terminal sushi/SCR domains plus the final LNR3 repeat and the two LNRs (LNR1 and LNR2) that are inserted into the catalytic domain. I understand that the four C-terminal domains may be missing due to intrinsic flexibility of this very large multidomain protein but it is not immediately obvious to me why LNR1/2 is missing. In the prediction by AlphaFold provided by UP it appears as a rigid appendix of the catalytic domain.

-Out of simple curiosity, could the authors please clarify, which the contribution of the DeepMind authors was that led to a model that was better than the automatically generated and currently available one from UniProt?

-The PDB Validation Report indicates many bond-angle outliers. Could these not be corrected during refinement by tighter restraints? The same holds for several atom clashes, which indicate that the

involved atoms are certainly too close. In contrast, there are no Ramachandran outliers and nearly no non-rotameric side chains.

-Page 4, line 156 (P4L156): please, replace apo-PAPP-A with unbound or substrate-unbound PAPP-A, etc. In enzymology, enzymes that require the aid of cofactors such as metal ions or prosthetic groups (e.g. a heme group, etc.) are holo-enzymes. In the absence of these groups, the enzyme is non-functional and is termed apo-enzyme. A common mistake that can be detected lately in the literature is to call a holo-enzyme that lacks a bound substrate or product an apo-form. Please, replace apo with unbound or similar throughout the text except when dealing with forms lacking the catalytic zinc ion, which are actually not dealt with in this manuscript.

-Given that, in addition to IGFBP-5, also IGFBP-2 and -4 are physiologically cleaved but lack the anchor helix of the former, how are these cleavages supposed to occur? When replacing the upstream cleavage-site segment of IGFBP-5 with that of IGFBP-4, the resulting protein is not cleaved (Fig. 3d, right panel). Have the authors assessed if this segment in IGFBP-2 and -4 adopts a helix as in IGFBP-5? Maybe recognition is topology-based and not sequence-based. It is difficult to conceive that such a super-specific peptidase has radically different mechanisms for cleaving just three substrates.

Reviewer #3 (Remarks to the Author):

This manuscript reports the structures of apo PAPP-A and PAPP-A bound to IGFBP-5. The structures reveal PAPP-A as a dimer arranged in a trans crossed over conformation (although a cis arrangement was not excluded), with one monomer being well defined and the other being poorly defined due to the apparent mobility of the two monomers relative to each other. This mobility made solving the structure challenging and a key approach to solving these structures involved the use of AlphaFold, an exciting computational method for predicting protein structures which predicts structures with high accuracy.

I am unable to provide detailed critique of the cryo-EM data analysis and computational methodology utilised as this is not my area of expertise. However, based on the information provided it is clear that the structures provide sufficient resolution in key areas to be able to make insightful observations relating to PAPP-A function as an IGFBP-5 protease. Revealed by the PAPP-A:IGFBP-5 structure and confirmed using PAPP-A mutants are the determinants of inter chain contact contributing to stability of the dimer. Also, the site of IGFBP-5 binding and cleavage is revealed. Interestingly, the only part of IGFBP-5 evident in the structure of the complex is a small helical fragment which includes the cleavage site adjacent to the coordinated zinc in the active site. Using a synthetic peptide equivalent to this fragment its affinity for PAPP-A was determined. A key determinant of substrate specificity previously identified in biochemical studies was also confirmed by this structure and using assays using a mutant

peptide. A mechanism for inhibition of PAPP-A by proMBP is proposed based on the location of a critical cysteine residue proximal to the IGFBP-5 binding site. Overall this study provides a major contribution to the understanding of the mechanism of action of this family of enzymes.

Comments/corrections:

The structures of the complex apparently have an IGFBP-5 119-143 fragment bound in each monomer active site. Is this consistent with known stoichiometry? Is it expected that two molecules of IGFBP-5 can be cleaved by the dimer?

The mutant PAPP-A constructs showed that dimerization is indeed required for effective IGFBP-4 cleavage. However, there is no speculation as to why a dimer is required. Has this anything to do with stabilisation of the substrate binding region? Is the stoichiometry of binding different for IGFBP-4?

From the PAPP-A structure and the knowledge of the substrate binding site revealed by the IGFBP-5 peptide interaction can you predict how the specificity of IGFBP-4 cleavage is achieved? The previous study by Laursen identified two arginines(126, 128) that are apparently important for substrate recognition. Can you model the IGFBP-4 equivalent region into the binding site and predict determinants of IGFBP-4 cleavage specificity?

For clarity it would be good in the introduction to acknowledge that there are other proteases that can cleave IGFBPs.

My understanding is that the earlier version of the computational method to predict structures was named AlphaFold2 whereas the latest version described in Jumper et al is termed AlphaFold. Which version was used in this study? The use of the nomenclature changes in this manuscript.

Is the affinity of the IGFBP-5 119-143 peptide similar to affinity for full length IGFBP-5?

Figure 1 legend: It would be good to define all domains. Missing are definitions for CCP, SCR and LNR

Line 107: "Only a helical peptide of IGFBP5 encompassing residues from 119 (NOT 129) to 143 was observed in the structure"

Line 607 Binding of IGFBP5 anchor peptide

Extended Figure 1: It would be good to state in the legend that the number on this figure starts at the signal peptide for each and that the numbering above in the domain labels is without the signal peptide. It would also be helpful to indicate where the sites of truncation in the mutants are located eg residue 1132 and residue 1267. Similarly it would be good to indicate Residue E483.

Extended Fig 8 has the construct design – it would be good to refer to this diagram early in the manuscript when first introducing the use of these.

Extended Fig 9: It would be good to annotate the IGFBP-5 with residue numbers – at least the N-terminal end

Extended Fig 12: The phylogenetic tree is not referred to anywhere in the text. The figure legend should include explanation of what the orange highlighted text is indicating. The statement in the abstract mentions that the 25 amino acid anchor peptide is not found in other IGFBPs and yet there is sequence similarity with IGFBP-3 (highlighted in orange). Also please explain the significance of the yellow highlighted lysine within the orange sequences. What do the black and grey boxes signify? It would also be good to highlight the IGFBP4 and IGFBP5 residues M135/K136 and S143/K144 respectively on the sequences

Response to the Reviewers' comments

We thank the reviewers for their recognition of the important biological questions addressed herein, and for their constructive feedback on the manuscript. In particular, we have included the structure of substrate-unbound PAPP-A, and also performed additional biochemical assays to address questions from the reviewers. Our point-by-point response to each question follows below.

Reviewer #1

RA Judge and colleagues report the structure of the metalloprotease PAPP-A protein in complex with its substrate IGFBP5 in attempt to delineate the mechanism of substrate recognition by PAPP-A. The study potentially addresses an interesting question regarding how PAPP-A selectively recognizes IGFBP5. However, the current set of structural and biochemical data run in a deficient way that may require additional clarification. Particularly, the lack of comparison with a substrate-free wild-type PAPP-A structure, and associated functional tests, leave it unclear whether the presented structure informs the exact mechanism of substrate selection or just provides one possible mechanism that is not sufficiently examined. It remains unclear if any conformational changes are required for or induced by the substrate recognition, which might be a crucial factor contributing to the substrate selectivity. This is worsened by the uncertainty between two possible models of cis against trans-dimer. Although this study looks like having some potential to go deeper and be conclusive, the authors appear to stop with a limited amount of data that provide less certainty for the underlying mechanism and the true structural model. Thus, I don't see that the current presentation meets the high quality commonly observed in a paper published in Nature Communications. Therefore, I would like to delay my recommendation until the authors answer the following enlisted questions, given that the authors might improve the manuscript to reach a more convincing conclusion and insights rather than leaving too many open questions around a very specific area.

(1) What is the structure of substrate-free wild-type PAPP-A? Is it still forming trans-dimer? How different is it as compared to the structure of IGFBP5-bound PAPP-A mutant? Any allosteric effects are involved in the substrate recognition? Any conformational changes are induced by substrate binding and contribute to substrate selectivity?

Great question. We did attempt to solve the structure of substrate-free WT PAPP-A. WT PAPP-A however has autocleavage issues during expression and purification leading to heterogeneity of the sample (**Supplementary Fig. 2a**), and the substrate-free protein sample showed greater flexibility compared with PAPP-ABP5, which prevented high resolution structure determination. Although the cryo-EM density map for WT protein has limited resolution ($<4.28 \text{ \AA}$), we could observe that the overall domain architecture is similar to that of PAPP-ABP5 and that of substrate-unbound PAPP-A (E483A), with all the datasets showing the same *trans*-dimer configuration (**Rebuttal Letter Fig. 1, Rebuttal Letter Fig. 2**). This indicates that there is no significant movement of the core domains upon binding of IGFBP5 (**Rebuttal Letter Fig. 2**). While we were able to make comparison between the WT map and the PAPP-ABP5 structure at a high level, the low resolution of the WT map did not support confident fitting of the protein backbone or protein side chains to allow determination of the structure at the atomic level, so we cannot comment on potential differences at the amino acid level. Additionally, as shown in the PAPP-ABP5 structure, the Zinc²⁺ coordination site is intact (**Fig. 3a, Supplementary Fig. 11f__new panel included in revision**), we therefore infer the E483A mutant may not introduce significant conformational changes. We have included some text in the 'PAPP-A substrate recognition of IGFBP5' and 'Discussion' section to clarify this (lines 195-202, and 335-340).

In the meantime, we were able to determine the structure of the 3.35 Å substrate-unbound PAPP-A (E483A), as this catalytically inactive protein does not have an autocleavage issue and exhibited a higher quality cryo-EM map (**Supplementary Fig. 4e-h, Supplementary Fig. 10_new figure included in revision**). This structure allows us to compare the effect of substrate binding which we will discuss in the next comment.

(2) How is the AlphaFold2-predicted PAPP-A compared to the IGFBP5-bound and substrate-free PAPP-A? How different are they? Any discussion on the implications of the difference might be considered? It would be nice to present the comparison in superimposed models.

This is a good suggestion. In our previous manuscript, we mentioned the cryo-EM map of substrate-unbound PAPP-A that was determined to $\sim 3.35 \text{ \AA}$. During paper revision, we further solved the structure of this substrate-unbound PAPP-A and included it in the revised manuscript (**Supplementary Fig. 10, Extended Table 2_modified table during revision**). It forms the same *trans*-dimer configuration and

similar to the PAPP-ABP5 structure, we see better density for one monomer than the other. As such we were able to reconstruct the full model for chain B but were only able to fit a partial model for chain A, while the chain B structure could enable a sufficient structural comparison. The substrate-unbound PAPP-A (E483A) and substrate-bound PAPP-ABP5 structures largely agree with each other (**Supplementary Fig. 12_new figure included in revision**), and no significant conformational changes were observed from the structure alignment either in the overall architecture or in the IGFBP5 binding groove (**Supplementary Fig. 12**). It's worth noting that multi-body refinement suggested that the two PAPP-A monomers in the substrate-unbound model have a greater twist relative to each other compared with PAPP-ABP5, suggesting substrate association reduces monomer dynamics in the PAPP-A *trans* dimer (**Extended Movie 1, Extended Movie 2_new movie included in revision**). We included the comparison of the structures with appropriate description in the section '**PAPP-A substrate recognition of IGFBP5**' (lines 164-173, and lines 188-202).

As the substrate-unbound and substrate-bound PAPP-A (E483A) largely agree with each other, we therefore used the PAPP-ABP5 structure to compare with the AlphaFold2 predicted model. The N-terminal core regions including the LG domain, MP domain and M1 domain superimpose well with quite low R.M.S.D values (**Supplementary Fig. 8_new figure included in revision**). M2 domain shows a bigger difference with a slightly higher R.M.S.D. Note M2 includes the dimerization interface hinge region and this region has a low confidence score in the prediction model. The CCP1/2 domains in the predicted model also largely agree with our experimental cryo-EM structure (**Supplementary Fig. 8**). The AlphaFold2 prediction thus was very helpful in solving the structure as it significantly reduced manual building of the individual domains. New description of this structure alignment was included in '**Structure of the PAPP-ABP5 complex**' part (lines 115-117) and in the figure legend section (lines 1104-1117).

The AlphaFold2 prediction does show a *cis* confirmation while the structure shows a *trans*-dimer, but we should not over interpret the AlphaFold2 prediction for monomer PAPP-A. It's not our intention that the monomer prediction should be used to determine the dimerisation mechanism of PAPP-A, especially experimental evidence is available. We have included some text describing this in the '**Discussion**' (lines 324-332). All legends for new figures are highlighted in the text.

(3) How much the E483A mutation effects on the conformation of PAPP-A? Does it keep the native structure of PAPP-A or induce certain structural rearrangement? It would be no help to take it as a given for a mutant structure and interpret it as if it is a wild-type structure. This creates more open questions than answers them.

As described above we did attempt to obtain a structure of WT-PAPP-A, but could only get a limited nominal resolution of $<4.28 \text{ \AA}$ cryo-EM map. **Rebuttal Letter Fig. 2** showed that the overall domain architecture of WT PAPP-A appears to be quite similar with PAPP-A E483A, but the low resolution does not permit us to do structural alignment at atomic level. To avoid auto-cleavage we therefore used the inactivating mutant (E483A) for structural studies. This is a common practice in cases of proteolytic activity and generally there is little difference observed between the WT and inactive mutants in terms of overall architecture. The Zinc²⁺ coordination site remains intact could also support the hypothesis that this catalytically inactive mutant should not introduce large conformational changes. While there may be differences at the atomic level while we cannot assess based on the current information we have, but we don't think this will impact our overall conclusion.

(4) If E483A mutation does induce structural changes, as expected for many enzymes or proteins, then the authors need to discuss its implication in substrate recognition. It may be necessary to point out the caveat of interpreting such a mutant structure in terms of substrate selectivity.

As described in answers for questions 1-3, we do not expect significant structural changes due to the catalytically inactive mutation. However, we could not rule out the possibility of subtle differences in the substrate binding site introduced by the E483A mutation. We have included some text in the '**PAPP-A substrate recognition**' and '**Discussion**' section to clarify this (lines 195-198, and 335-348).

(5) The domain swap architecture is interesting. Is this architecture also preserved in the wildtype PAPP-A dimer structure? How certain is the domain swap architecture, given the lower resolution in the neck region of the swap domains (shown in Fig. 1b and Fig. 2a)? Fig. 2a appears to contradict Supplementary Fig. 6. No better explanation and solid evidence why the authors choose the domain-swapped model? The data presentation or discussion do not appear to be sufficiently convincing in the regard.

Although the WT-PAPP-A dataset has quite limited resolution, we do observe the *trans* dimer for the WT protein (**Rebuttal Letter Fig 2**) as we also do for the substrate unbound PAPP-A (E483A) ($\sim 3.35 \text{ \AA}$) and substrate-bound PAPP-A (E483A)_{BP5} ($\sim 3.28 \text{ \AA}$) structures. We took your suggestion and tried a few different methods to obtain a better reconstruction for the neck region of the map. This included focused classification on the neck region with and without alignment in Relion 4, multi-body analysis and subsequent examination of eigenvector subsets and 3DVA cluster analysis/refinement in CryosparcV3. Unfortunately, the continuous motion

between the two halves did not allow us to improve the map in this area, and we decided to stick to the data shown in the manuscript. It's quite clear in the cryo-EM map that the domains swop over and form the *trans* dimer. Some of the language we used in the last version of manuscript may have diluted this and we have strengthened our description of the *trans*-dimer observation. Part of the confusion is due to the AlphaFold prediction (**Supplementary Fig. 6**) being in the *cis* conformation. We are confident in the *trans* dimer shown in the map and would caution against over interpretation of how the AlphaFold monomer prediction might apply to a higher order assembly. We have modified in the text in the '**PAPP-A substrate recognition of IGFBP5**' and '**Discussion**' sections (lines 164-173, and lines 324-335).

(6) The authors need to show the Zinc density and the side chains at the active site, as a direct validation of the state of active site. What about the validation using FSC to measure how good the atomic model fitting the cryo-EM map? How valid is the currently reported 3.3 Å overall resolution? The local-resolution map of the apo mutant PAPP-A should also be shown.

Thank you for the suggestion. We previously highlighted the zinc ion and side chains at the active site (**Fig. 3a**), and we further included an enlarged view of the active site in the **Supplementary Fig. 11f** (*_new figure included in revision*), and matched side chains for S143/H486/H482/H492 in the cryo-EM density map (shown in mesh density). The density map shows a continuous envelop around the residues and the Zn²⁺ in the active site.

As suggested, we further included the local resolution map for the substrate-unbound PAPP-A (E483A) in **Supplementary Fig. 10a**. The reported resolution for both structures (Substrate-unbound and substrate-bound PAPP-A) is based on the gold standard of FSC=0.143 criteria. Although resolution varies across the substrate-bound PAPP-A_{BP5} and substrate-unbound PAPP-A (E483A) map, the metalloprotease region was showed to have a high resolution, which allowed us to confidently build the model of the active site.

(7) How about the prediction of the dimer structure by AlphaFold-Multimer? Does it also predict a *trans*-swapped dimer model or still a *cis*-dimer? What about the wild-type PAPP-A dimer structure? How much is the *cis/trans* confusion affected by the E483A mutation?

Firstly, we would like to clarify that when this study took place only a monomeric AlphaFold2 prediction was available. Therefore the utility of AlphaFold2 model lay in predicting the structures of the novel domains as well as the core domain packing, with sufficient accuracy to aid in density interpretation and model-building. While we mention in passing that the AlphaFold2 prediction was more consistent with a *cis*-dimer, it is not our intention that a monomeric prediction should be used to decide this question. Even AlphaFold Multimer has not been rigorously evaluated on dimers of this size, nor on a test set composed mainly of swapped domains. Therefore, given that experimental data is available in the form of the Cryo-EM maps, we are more inclined to rely on that data than on the AlphaFold2 prediction in this case.

Nevertheless, we have run PAPP-A through AlphaFold Multimer, which resulted in a *cis*-dimer prediction the majority of the time (although a *trans*-dimer was produced in one specific setup described below in **Rebuttal Letter Fig. 3**). In more detail: we ran the 5 recently-released AlphaFold Multimer models (v2.2.0). Initially a truncated sequence was used as input to the Colab, consisting of PAPP-A residues 90-316 and 967-1344 (numbering relative to the UniProt entry Q13219). This truncation aimed to preserve the most relevant domains, while avoiding out of memory errors and keeping the input length in the range for which the model has been validated. Four out of the five predictions were *cis*-dimer, while the least confident was a *trans*-dimer (ranking based on weighted pTM / ipTM average as described in Evans et al). Repeating this experiment for a more complete PAPP-A dimer consisting of residues 90-1344 yielded only *cis*-dimer predictions, and introducing an E563A mutation into this sequence gave the same result.

Moreover, as mentioned in the answer to question #1, the 4.28 Å WT PAPP-A map also shows a *trans* dimer configuration. The active site residue E483 is far away from the dimer interface so we hypothesize that it should not affect the dimer conformation. Again we want to emphasize that while the AlphaFold2 predictions were valuable for fitting the domains the experimental maps show a *trans* dimer and we give preference to the experimental data over the predictions.

(8) Most importantly, due to the modeling uncertainty and lack of a wildtype structure comparison, how much the reported complex represents a functional complex is a major open question that is central to the subject of this study? Without addressing this issue to a sufficient level, it may not be a good fit for a journal like Nature Communications, where readers are expecting more certain answers rather than just tentative models with some major uncertainty.

Thanks for the question. Combined with the new substrate-unbound PAPP-A (E483A) structure, we think our data together showed certainty for the following points: 1) Although the WT-PAPP-A cryo-EM density map is of quite limited resolution, it does show the *trans* dimer and outlines the position of sub-domains. In fact, all the experimental maps/structures (WT, substrate-unbound and substrate-bound) show a clear *trans* dimer. 2) The structure for the substrate-unbound PAPP-A (E483A) which we have included in the revised manuscript allowed comparison to the PAPP-A_{BP5} complex structure. Here we did not observe any significant conformational changes between the two structures. 3) As far as we can determine there is no significant movement of domains between the structures/maps of these three entities. Combined with our biochemical functional analysis to understand the substrate recognition and selectivity mechanism, we think the reported PAPP-A_{BP5} structure represents a functional complex.

(9) Both the title and abstract did not mention that the structure is analyzed with mutant PAPP-A, with considerable uncertainty in the dimer architecture, which felt a little bit deceptive. The fact that AlphaFold2 did not help clarify this, but instead, make the uncertainty greater by predicting cis-dimer, leaves it somewhat ironic, in contrary to the rhetoric in the abstract that emphasizing “by leveraging the power of AlphaFold2”. If the authors cannot fully conclude on trans-dimer against cis-dimer, why make such a conclusion in the abstract and claiming that they have solved the structure of trans-dimer? I think the right way to handle this is to improve the local resolution of the neck region of the density map by either adding more data or by better 3D classification approach. Eventually, a high-resolution cryo-EM density at this critical local region itself presents the most interpretable data and evidence to solve the modeling ambiguity.

As aforementioned in the answer to question #7, the utility of AlphaFold2 lay in predicting the structures of the novel domains as well as the core domain packing, with sufficient accuracy to aid in density interpretation and model-building, and it is not our intention that a monomeric prediction should be used to decide the question of the configuration of the dimer. So we should not overinterpret the *cis* conformation predicted by AlphaFold2. In the answer to question #5, we explained our attempts to improve neck region resolution, but the continuous motion between the two halves did not allow us to improve the map in this area. Nevertheless, the PAPP-A_{BP5} map clearly shows a *trans* dimer conformation. As aforementioned, we modified our text in the ‘**PAPP-A substrate recognition of IGFBP5**’ and ‘**Discussion**’ sections (lines 164-173, and lines 324-335) . We change the ‘**Title**’ from PAPP-A_IGFBP5 complex to PAPP-A_{BP5}. We think it could be better not to demonstrate E483A in title as this detail will be reflected all across the manuscript. Based on reviewer’s suggestion, we also modified our ‘**Abstract**’ to

emphasize the fact that the structures were analyzed with the catalytically inactive PAPP-A (line 25).

My last comment is that to solidify the mechanism of dynamic interactions between protein counterparts often requires determination of multiple conformations at representative functional states/stages. For PAPP-A and IGFBP5, this does not seem to be an exception. Limited by insufficient resolution on the critical local region and inconsistency between human interpretation and AlphaFold2 prediction, this study opens more questions than it answers. Most problematically, the final structural model is unfortunately uncertain, leaving the mechanistic interpretation too speculative and preliminary to publish in any serious research journal. The authors really need put on some efforts to at least remove the structural ambiguity before deciding on final publication.

We thank the reviewer for the comments and suggestions. The work we have done in response to these comments (including the substrate-unbound structure) has certainly strengthened, clarified and made the manuscript better. Our cryo-EM structures are clear and we further modified our language to strengthen these experimental observations (eg, *trans*-dimerization). We think our responses to the prior above comments addressed the reviewer's summary statement.

Reviewer #2:

The manuscript by Judge and colleagues reports on structural studies by cryo-EM aimed at deciphering the mechanism employed by the metalloendopeptidase PAPP-A to selectively cleave its highly specific substrates, viz. three out of six insulin-like growth factor binding proteins (IGFBP-2, -4 and -5). Given that the enzyme does not cleave any other substrates, it has been long speculated which the molecular determinants underlying this very high specificity would be. Here, Judge and colleagues, assisted by AlphaFold-frontman John Jumper, come up shedding light on the solution with a partial structure of the enzyme in complex with a short helical fragment of substrate IGFBP-5, i.e. a product complex. This is a very interesting and long-awaited structure, which clearly should deserve publication in Nature Communications. However, there are a series of issues that I would need the authors to clarify/amend before definitive acceptance.

-The reported structure is a product complex of most of PAPP-A with a very small fragment of IGFBP-5 (25 residues), i.e. not the entire proteins. Please, make this clear in the title and throughout the text.

Thanks for the reviewer's suggestion. When performing the cryo-EM analysis for the complex, we used full length PAPP-A with the catalytically inactive mutation (to avoid auto cleavage or cleavage of the substrate) and full length IGFBP5. Due to the flexibility of the complex, the cryo-EM structure does not reveal the extended C-terminal structure of PAPP-A nor the majority of IGFBP5 which is disordered in the structure. Only the IGFBP5 anchor peptide is clearly observed. We describe and discuss the aforementioned throughout the manuscript. Following the reviewer's suggestion to make this pointer clearer, we further changed the **Title** from 'PAPP-A_IGFBP5 complex' to 'PAPP-A_{BP5}' (line 1) to highlight the fact that only a small peptide of IGFBP5 was observed. We also further emphasized this in '**Structure of PAPP-A_{BP5} complex**' session (lines 112-115), and the name (PAPP-A_{BP5}) was used all through the revised manuscript.

-Please, adopt the numbering of the corresponding UniProt entry (UP Q13219) for clarity. The current numbering lags 80 positions behind, which is misleading. The same holds for IGFBP-5. It is very useful to have a homogeneous numbering throughout the distinct databases.

Thanks for raising this point. PAPP-A has a 80-residues long signal sequence that is present in the UniProt numbering which is cleaved off in the secreted protein. In this manuscript, we chose to number the protein sequence starting after the removal of the signal sequence so as to match the referenced literature reports. For example, in some previous reports (References 11, 22, 24, 28, 31, 38, 41), the authors described residues of PAPP-A that play key roles in the protein function. We think keeping the same numbering makes it easier to compare our structure with the findings in these reports. The same decision was made regarding IGFBP5. IGFBP5 has a 20 residues-long signal peptide which is present in the UniProt numbering which is also naturally cleaved off during protein production. In the previous literature reports (References 36, 42), the authors numbered IGFBP5 starting after removal of the signal peptide. As we discuss various key residues in our manuscript that were reported by those previous authors, to avoid confusion, we thought it best to use sequence numbering consistent with that found in these literature reports. We have included text in the **'Methods'** session (line 511, lines 679-685), and figure legends for **Supplementary Fig. 1** (line 1018) and **Supplementary Fig. 15** (line 1217).

-The cryo-EM map shows clear density for most of the N-linked glycosylations. For the sake of model completeness, I would invite the authors to model them, even if only for one NAG unit.

Thanks for the suggestion. In our final structures, we do see some varied densities at various glycosylation sites that have been previously identified in the literature (Reference 22) or Uniprot (Q13219), but the densities at these sites were poorly defined (**Rebuttal Letter Fig. 4**) and the attempts to fit even one NAG at these sites gave us poor results in refinement. We therefore did not include sugars in the final structure model. Following the suggestion, we added a description in the **'Structure of the PAPP-A_{BP5} complex'** section (lines 108-109).

-The structure encompasses nine domains from the entire structure, all except the three C-terminal sushi/SCR domains plus the final LNR3 repeat and the two LNRs (LNR1 and LNR2) that are inserted into the catalytic domain. I understand that the four C-terminal domains may be missing due to intrinsic flexibility of this very large multidomain protein but it is not immediately obvious to me why LNR1/2 is missing. In the prediction by AlphaFold provided by UP it appears as a rigid appendix of the catalytic domain.

Unfortunately the map in the region of the LNR1 and LNR2 domains is not well defined. We fit what we could confidently fit in that region. The lack of density indicates that there is flexibility in these domains that reduced the signal to noise in this area and we think this is due to the complexation with IGFBP5. Previous reports (Reference 38, 42) suggested that all three LNR domains function together and are strictly required for substrate selectivity for IGFBP4 but not for IGFBP5. Perhaps this is why the LNR domains are disordered and not observed in our IGFBP5 complex structure. Our biochemical data in **Fig. 4** showed PAPP-A has different recognition mechanism towards IGFBP4 and IGFBP5. To further understand whether LNR1/2 could interact with LNR3 to form the LNR center, we performed molecular dynamic simulations as a complementary method to the experimental cryo-EM data. The MD-simulation result suggested that in the context of IGFBP5 association, it's energetically unfavored to form this LNR center (**Supplementary Fig. 19_new figure included in revision**). We included the discussion for MD-simulation in the '**PAPP-A substrate selectivity**' part (lines 285-301). The MD-simulation result in turn supported that IGFBP5 recognition is LNR1/2-LNR3 interaction independent. Perhaps what is needed to see the LNR centers (LNR1 and LNR2 in complex with LNR3) is a structure of the complex of PAPP-A/IGFBP4/IGF. As discussed in the '**Conclusion**' part, the structure of PAPP-A/IGFBP4/IGF1 is highly desired, but it is beyond the scope of this current effort and is the topic of further work. New method section was included in line 777-847.

With regard to AlphaFold2, the low AlphaFold pLDDT performs well as a predictor of disorder. This does not rule out the possibility that a high confidence domain could contain flexible regions or adopt alternative conformations under certain conditions.

-Out of simple curiosity, could the authors please clarify, which the contribution of the DeepMind authors was that led to a model that was better than the automatically generated and currently available one from UniProt?

The contribution of the DeepMind authors consisted of generating predictions (prior to AlphaFold open-sourcing) and assisting in their interpretation. The monomeric prediction used in this study is not expected to be any better than the one displayed in UniProt, and in fact differs only in minor ways like the position of the unstructured signal peptide.

-The PDB Validation Report indicates many bond-angle outliers. Could these not be corrected during refinement by tighter restraints? The same holds for several atom clashes, which indicate that the involved atoms are certainly too close. In contrast, there are no Ramachandran outliers and nearly no non-rotameric side chains.

The resolution of PAPP-_{ABP5} map varies widely across the structure, and some local regions have quite limited resolution. The protein in many places is also tightly packed. The low resolution and tight packing make it difficult to reduce some of the atom clashes and bond-angle outliers. Considerable time was spent in refining the structure and we feel that given the limitations of the data further efforts would take more time while providing only minimal improvement, so no further changes were made in this regard.

-Page 4, line 156 (P4L156): please, replace apo-PAPP-A with unbound or substrate-unbound PAPP-A, etc. In enzymology, enzymes that require the aid of cofactors such as metal ions or prosthetic groups (e.g. a heme group, etc.) are holo-enzymes. In the absence of these groups, the enzyme is non-functional and is termed apo-enzyme. A common mistake that can be detected lately in the literature is to call a holo-enzyme that lacks a bound substrate or product an apo-form. Please, replace apo with unbound or similar throughout the text except when dealing with forms lacking the catalytic zinc ion, which are actually not dealt with in this manuscript.

This is an excellent point! Following the suggestion, we made changes across the abstract whole text, method, and all figure legends to replace apo-PAPP-A with substrate-unbound PAPP-A. We also clarified Wild type protein (WT) where listed. As too many changes have been made regarding this (highlighted in text), we therefore didn't list all line numbers here.

-Given that, in addition to IGFBP-5, also IGFBP-2 and -4 are physiologically cleaved but lack the anchor helix of the former, how are these cleavages supposed to occur? When replacing the upstream cleavage-site segment of IGFBP-5 with that of IGFBP-4, the resulting protein is not cleaved (Fig. 3d, right panel). Have the authors assessed if this segment in IGFBP-2 and -4 adopts a helix as in IGFBP-5? Maybe recognition is topology-based and not sequence-based. It is difficult to conceive that such a super-specific peptidase has radically different mechanisms for cleaving just three substrates.

This is a great question. We carefully examined IGFBP2 and IGFBP4 cleavage and confirmed that their cleavage is IGF-dependent, which is distinct from IGFBP5

(Rebuttal Letter Fig. 5). Our hypothesis is that IGF binding will introduce conformational changes to IGFBP2 and IGFBP4 and that the formed complex will then be in a favored conformation for PAPP-A recognition. Looking at the AlphaFold2 predicted IGFBP2 and IGFBP4 structures, they also reveal a helical portion in the corresponding area as the IGFBP5 anchor peptide with the regions flanking the helical peptide predicted as flexible loops (**Rebuttal Letter Fig. 6**). However, the length of the IGFBP2/4 helical peptides are shorter compared with IGFBP5 and we simply do not know how IGF binding effects the structure of IGFBP2 or 4.

Our biochemical data revealed that the chimeric IGFBP5 could not be efficiently recognized by PAPP-A (**Fig. 3e**), suggesting although IGFBP4 showed the helical conformation in the corresponding IGFBP5 anchor region, this helix will not behave exactly as the IGFBP5 anchor peptide. This hypothesis is further confirmed by analyzing the AF568 labeled IGFBP4¹¹⁴⁻¹³⁴ peptide interaction with PAPP-A in the FP assay. As showed in the **Rebuttal Letter Fig. 7**, the IGFBP4¹¹³⁻¹³⁴ peptide has a much lower binding affinity with PAPP-A (KD measured ~ 6 μ M) compared with IGFBP5 anchor peptide (KD measured at 280nM). Taken together, we think this unique IGFBP5 anchor peptide provides a specific and strong binding with PAPP-A, and therefore it could achieve an efficient cleavage independent of IGFs (EC₅₀ ~53pM as shown in **Fig. 3c**). The structure of a PAPP-A/IGFBP4/IGF1 complex would be highly desired to understand this mechanism but is beyond the scope of this work.

Reviewer #3

This manuscript reports the structures of apo PAPP-A and PAPP-A bound to IGFBP-5. The structures reveal PAPP-A as a dimer arranged in a trans crossed over conformation (although a cis arrangement was not excluded), with one monomer being well defined and the other being poorly defined due to the apparent mobility of the two monomers relative to each other. This mobility made solving the structure challenging and a key approach to solving these structures involved the use of AlphaFold, an exciting computational method for predicting protein structures which predicts structures with high accuracy.

I am unable to provide detailed critique of the cryo-EM data analysis and computational methodology utilised as this is not my area of expertise. However, based on the information provided it is clear that the structures provide sufficient resolution in key areas to be able to make insightful observations relating to PAPP-A function as an IGFBP-5 protease. Revealed by the PAPP-A:IGFBP-5 structure and confirmed using PAPP-A mutants are the determinants of inter chain contact contributing to stability of the dimer. Also, the site of IGFBP-5 binding and cleavage is revealed. Interestingly, the only part of IGFBP-5 evident in the structure of the complex is a small helical fragment which includes the cleavage site adjacent to the coordinated zinc in the active site. Using a synthetic peptide equivalent to this fragment its affinity for PAPP-A was determined. A key determinant of substrate specificity previously identified in biochemical studies was also confirmed by this structure and using assays using a mutant peptide. A mechanism for inhibition of PAPP-A by proMBP is proposed based on the location of a critical cysteine residue proximal to the IGFBP-5 binding site. Overall, this study provides a major contribution to the understanding of the mechanism of action of this family of enzymes.

Comments/corrections:

The structures of the complex apparently have an IGFBP-5 119-143 fragment bound in each monomer active site. Is this consistent with known stoichiometry? Is it expected that two molecules of IGFBP-5 can be cleaved by the dimer?

In this study, we used FL IGFBP5 for structural analysis and the cryo-EM map clearly revealed one IGFBP5 anchor peptide density in each PAPP-A monomer, so we

conclude that two IGFBP5 molecules can be recognized and cleaved by a PAPP-A homodimer. With regarding to using biochemical assays to understand stoichiometry, we attempted a SEC-MALS assay using the IGFBP5 anchor peptide to analyze complex molecular weight, however, the result was inconclusive as the difference of PAPP-A_{BP5} compared with PAPP-A alone was too subtle to allow us draw a conclusion.

The mutant PAPP-A constructs showed that dimerization is indeed required for effective IGFBP-4 cleavage. However, there is no speculation as to why a dimer is required. Has this anything to do with stabilisation of the substrate binding region? Is the stoichiometry of binding different for IGFBP-4?

This is a great question. One of the key differences here is that PAPP-A cleavage of IGFBP4 is IGF dependent while PAPP-A cleavage of IGFBP5 is not (**Rebuttal Letter Fig. 5**). Unfortunately, there are no publicly available structures showing how binding of IGF influences the conformation of IGFBP4 which in turn will influence PAPP-A recognition. Based on the information we have gained in this study, we hypothesize that the IGF/IGFBP4 complex binds to the extended *trans* C-terminal portion of the dimer (including CCP1 where S1144Y is located), across the dimer interface and also to the protease domain to facilitate effective cleavage. In this way two IGFBP4/IGF complexes could be bound by one PAPP-A homo-dimer. This is currently a hypothesis however as our current structural and biochemical information does not provide direct insights regarding how the IGFBP4/IGF complex interacts with the dimer. For this we will need the ternary complex structure of PAPP-A/IGFBP1/IGF1. This however it is beyond the scope of this current effort and is the topic of further work. We included a sentence describing this hypothesis in the '**Discussion**' section (lines 347-353).

From the PAPP-A structure and the knowledge of the substrate binding site revealed by the IGFBP-5 peptide interaction can you predict how the specificity of IGFBP-4 cleavage is achieved? The previous study by Laursen identified two arginines(126, 128) that are apparently important for substrate recognition. Can you model the IGFBP-4 equivalent region into the binding site and predict determinants of IGFBP-4 cleavage specificity?

The issue in modeling IGFBP4 binding is that IGFBP4 cleavage is IGF dependent (as shown in **Rebuttal Letter Fig. 5**) and we simply do not know how IGF binding effects the conformation of IGFBP4. As shown in the **Rebuttal Letter Fig. 6**, R126 and R128 both reside on the IGFBP4 helix AlphaFold prediction that corresponds to the IGFBP5 anchor

peptide but we do not know how IGF binding effects this section of IGFBP4. Our biochemical data suggests this IGFBP4 helix is not as strong an anchoring point as it is for IGFBP5 (**Fig. 3e, Rebuttal Letter Fig. 7**). Moreover, we tested the IGFBP4 (R126A/R128A) double mutant in our gel-based proteolytic activity assay. As shown in Rebuttal Letter Figure. 8, the IGFBP4 mutant could be cleaved efficiently by PAPP-A, at a similar efficiency as WT IGFBP4. The previous result suggested that the mutant of the two arginines only slightly affect IGFBP4's recognition by PAPP-A (*Laurson et al.* Figure 3 in the paper), which actually agrees with our data. To really understand the functional mechanism of IGFBP4 recognition (in an IGF dependent manner) we will need to get the structure of the PAPP-A/IGFBP4/IGF complex, while this is currently beyond the scope of this study.

For clarity it would be good in the introduction to acknowledge that there are other proteases that can cleave IGFBPs.

The reviewer's suggestion is well noted. Text was added to the '**Abstract**' (line 22) and background introduction part (line 22 and 49).

My understanding is that the earlier version of the computational method to predict structures was named AlphaFold2 whereas the latest version described in Jumper et al is termed AlphaFold. Which version was used in this study? The use of the nomenclature changes in this manuscript.

This naming confusion is indeed unfortunate. DeepMind participated in CASP13 as team "AlphaFold" and in CASP14 with a completely new model as "AlphaFold2". Our intention was to drop the "2" from the name on publication of the CASP14 model in Jumper *et al.* However, in practice both names AlphaFold and AlphaFold2 are currently in use to refer to our CASP14 model.

I can confirm that the version of the model used in this study was the same one entered into CASP14; equivalent to our first publicly-released monomer model at <https://github.com/deepmind/alphafold>. The name AlphaFold2 has therefore been used in this manuscript.

Is the affinity of the IGFBP-5 119-143 peptide similar to affinity for full length IGFBP-5?

We further performed an fluorescence polarization interaction assay to measure FL IGFBP5 interaction with the catalytically inactive mutant PAPP-A (E483A). The IGFBP5 with Fam labeling was shown to bind PAPP-A at a slightly higher K_D (~250nM) compared to the IGFBP5¹¹⁹⁻¹⁴³ anchor peptide (K_D measured at 380nM). The similar binding affinity suggests the IGFBP5 anchor peptide is the primary contributor to PAPP-A binding. At the meanwhile, we cannot rule out the possibility for other interfaces between IGFBP5 and PAPP-A. The new figure was added in **Fig. 3c(*new figure included in revision*)** and text was expanded in lines 208-210. Figure legend was modified in lines 970-973. We also expanded the description in **Methods** section (lines 687-698, 720-729).

Figure 1 legend: It would be good to define all domains. Missing are definitions for CCP, SCR and LNR.

Thanks for the reviewer's suggestion. Descriptions for all the domains have been added to the **Figure legend** starting where they are first introduced in **Figure 1a** (lines 926-930).

Line 107: "Only a helical peptide of IGFBP5 encompassing residues from 119 (NOT 129) to 143 was observed in the structure" .

Thanks for catching the typo. Corrected text in line 113.

Line 607 Binding of IGFBP5 anchor peptide

Thank you, fixed the spelling in the text Line 702.

Extended Figure 1: It would be good to state in the legend that the number on this figure starts at the signal peptide for each and that the numbering above in the domain labels is without the signal peptide. It would also be helpful to indicate where the sites of truncation in the mutants are located eg residue 1132 and residue 1267. Similarly it would be good to indicate Residue E483.

Thanks for the suggestion. We stated that we numbered residues after the signal peptide in the **Supplementary Fig.1** legend (line 1018). We stated the way for numbering in the legend **Sup Fig. 15** (line 1217). Similar descriptions were further

included in 'Methods' section as well (lines 679-685). We highlighted the truncations used in the paper (PAPP-A (1132) and PAPP-A (1267)) in the sequence alignment diagram. We also highlighted the E483 as well as the 4-cysteine residues that are discussed in the manuscript. All the optimizations were reflected in the **Supplementary Fig. 1**. The figure legend was also optimized accordingly (lines 1015-1018).

Extended Fig 8 has the construct design – it would be good to refer to this diagram early in the manuscript when first introducing the use of these.

Thanks for the suggestion. We referred to this figure (Currently renumbered to **Supplementary Fig. 9**) at the first place we introduced the truncation/mutants (Line 133).

Extended Fig 9: It would be good to annotate the IGFBP-5 with residue numbers – at least the N-terminal end

Thanks for the suggestion. We further annotated the terminal residue in the figures (Renumbered to **Supplementary Fig. 11d, e, f**).

Extended Fig 12: The phylogenetic tree is not referred to anywhere in the text. The figure legend should include explanation of what the orange highlighted text is indicating. The statement in the abstract mentions that the 25 amino acid anchor peptide is not found in other IGFBPs and yet there is sequence similarity with IGFBP-3 (highlighted in orange). Also please explain the significance of the yellow highlighted lysine within the orange sequences. What do the black and grey boxes signify? It would also be good to highlight the IGFBP4 and IGFBP5 residues M135/K136 and S143/K144 respectively on the sequences

We thank the reviewer for noticing this, and we removed the phylogenetic tree as that is not referred anywhere in the text.

We described in the manuscript that the 25-amino acid anchor peptide is not found in other PAPP-A substrates (IGFBP2 and IGFBP4). IGFBP3 is the only homology that show relatively high sequence conservation with IGFBP5 anchor peptide, but it's not a PAPP-A substrate. Highlighted in yellow within the orange sequence is the anchoring

residue K128 in IGFBP5, and the yellow highlight for IGFBP3 was further removed as we did not examine the function of this residue in this study. The specific PAPP-A cleavage site residues of IGFBP4 and IGFBP5 were further highlighted in red, and the cleavage site is labeled by green arrow. All optimizations could be found in the current **Supplementary Fig. 15**, and the figure legend was optimization accordingly (lines 1213-1217).

Rebuttal Letter Fig. 1: Local resolution of WT PAPP-A cryo-EM density map

Cryo-EM density map of WT PAPP-A at an overall resolution lower than 4.28Å, with two threshold levels used to highlight local resolution (left: 0.0783 and right: 0.121). The density map is of low resolution and one monomer exhibits a significant higher resolution than the other. The 3D cryo-EM map is colored by resolution (bar in the middle). The local resolution is calculated by CryoSPARCv3.

Rebuttal Letter Fig. 2: Comparison of WT PAPP-A density map and PAPP-ABP5 structure

Left: Overlay of PAPP-ABP5 complex structure with WT PAPP-A cryo-EM density map (4.28 Å with contour level 0.085). Right: Angled view of the structure alignment. WT PAPP-A cryo-EM map shows a *trans*-dimer configuration. The PAPP-ABP5 structure is shown as cartoon, with monomer A in maroon and monomer B in light blue. WT PAPP-A cryo-EM map is shown in grey. IGFBP5 anchor peptide is highlighted in dark orange.

Rebuttal Letter Fig. 3: AlphaFold Multimer predicted dimer for a truncated PAPP-A sequence.

The input sequence for the predictions above consisted of residues 90-316 + 967-1344, a choice made to preserve the domains involved in dimerization while remaining within the length range for which AlphaFold Multimer has been validated. The most confident prediction was a *cis*-dimer (Top panel; predicted chains colored red and blue, with the AlphaFold Protein Structure Database prediction for PAPP-A aligned to each chain to produce the grey surface). One out of five models produced a *trans*-dimer prediction (Bottom panel). Predictions were generated using the AlphaFold Colab with models from release v2.2.0.

Rebuttal Letter Fig. 4: Extra densities in the glycosylation site of PAPP-A

In the PAPP-ABP5 cryo-EM structure, we observed extra densities in the reported glycosylation sites of PAPP-A, and a few examples were shown in the figure. However, those densities were poorly defined which prevents confident fitting of sugars.

Rebuttal Letter Fig. 5: Both IGFBP4 and IGFBP2 cleavage by PAPP-A are IGF-dependent

Gel-based proteolytic activity of WT PAPP-A towards IGFBP4 and IGFBP2 in the presence or absence of IGF1. IGFBP4 or IGFBP2 (400nM) were either pre-incubated with IGF1 (700nM) or no IGF1 at room temperature for 30 mins. The mixture were then incubated with WT PAPP-A in a serial dilution at 37°C for 4 hours. **Left**, PAPP-A was added in a 1.5-fold serial dilution for IGFBP4 cleavage. **Right**, PAPP-A was added in a 2-fold serial dilution for IGFBP2 cleavage. The result suggested the addition of IGF1 significantly increased IGFBP2 cleavage efficiency.

Rebuttal Letter Fig. 6: Analysis of the corresponding region of IGFBP4 and IGFBP2 with IGFBP5 anchor peptide

In order to assess the whether IGFBP2 and IGFBP4 also adopt helical conformation in the corresponding region of IGFBP5 anchor peptide, we leveraged the AlphaFold predicted models: IGFBP4 (<https://alphafold.ebi.ac.uk/entry/P22692>), IGFBP2 (<https://alphafold.ebi.ac.uk/entry/P18065>), and IGFBP5 (<https://alphafold.ebi.ac.uk/entry/P24593>). Interestingly, IGFBP2 and IGFBP4 also showed a helical conformation in this region in the predicted model, however with a shorter length compared with IGFBP5 anchor peptide.

Rebuttal Letter Fig. 7: IGFBP4¹¹³⁻¹³⁴ binding affinity analysis with PAPP-A

Binding affinity between the IGFBP4¹¹³⁻¹³⁴ and WT PAPP-A measured by fluorescence polarization (FP) assay. The N-terminal Alexa Fluor 568 conjugated IGFBP4¹¹³⁻¹³⁴ (5nM) was incubated with serial dilutions of WT PAPP-A (50μM to 6nM) in PBS buffer at room temperature. The FP values were recorded at 10min. Data shown is the representative of three replicates. The data were analyzed in GraphPad Prism 9.1.2.

Rebuttal Letter Fig. 8: PAPP-A could efficiently cleave IGFBP4 R126A/R128A, similar as WT IGFBP4

(Left) PAPP-A cleavage for WT IGFBP4. (Right) PAPP-A cleavage for IGFBP4 R126A/R128A double mutant. IGFBP4 or IGFBP4 mut (400nM) were either pre-incubated with IGF1 (700nM) or no IGF1 at room temperature for 30 mins. The mixture were then incubated with WT PAPP-A in a serial dilution at 37°C for 4 hours.

REVIEWERS' COMMENTS

Reviewer #1 (Remarks to the Author):

The authors have sufficiently addressed all my questions. I'd like to recommend its publication in the present form.

Reviewer #2 (Remarks to the Author):

The authors addressed my concerns adequately and the manuscript would be acceptable for publication in my view.

Minor points:

Legend to Rebuttal Letter Fig. 1: CryoSPARCv3.

Reviewer #3 (Remarks to the Author):

The manuscript revision now includes and is greatly improved by the low resolution structure of the substrate free PAPP-A mutant (E483A). The authors point out the difficulties in generating a structure of the wild type PAPP-A and acknowledge this in the manuscript. This addition has provided evidence to support that the unbound and bound enzyme is found as a dimer with similar dimerization mode ie trans.

The corrections add sufficient explanation of the possible mechanisms underlying the substrate recognition of IGFBP-4 versus IGFBP-5 and it is reasonable to suggest that having the complex of PAPP-A with IGF:IGFBP-4 is beyond the scope of this manuscript as that represents a significant body of work.

The addition of the fluorescence polarization assays for FL IGFBP-5 and the fragment has provided some clarity regarding the binding to PAPP-A and the authors are conservative in their conclusions with respect to the possibility of other interactions contributing to binding.

Since the original review of this manuscript a paper has been published describing the role of the linker domain in IGF-dependent protease cleavage of IGFBP-2 (Jaipuria et al Proteins DOI 10.1002/prot.26350). The conclusions are that "L-hIGFBP2 does not undergo a well-defined conformational change upon

binding its ligand, but binding is accompanied by a significant change in dynamics on both the millisecond-microsecond and picosecond-nanosecond time scales". They conclude it is the change in flexibility of the linker that increases the protease susceptibility. This information should be included in the current manuscript in the discussion of possible mechanisms of PAPP-A cleavage of IGFBP-4 and IGFBP-2. Furthermore, it would be good to acknowledge that it has not been possible to generate a structure of the IGFBP linker domain even when in complex with an IGFBP. Even in the recent cryo EM structure of IGF-I:IGFBP-3:ALS complex (Kim et al Nat Comm 2022) the linker domain remains flexible and very little density is seen for the linker domain. This information supports the suggestion that there is no significant gaining of structural stability upon ligand binding and would suggest that if there is any change in structure it is likely to be in a limited region rather than across the entire linker.

Response letter to reviewer's comments

Reviewer #1

The authors have sufficiently addressed all my questions. I'd like to recommend its publication in the present form.

We thank the reviewer's recognition for our work, and all the previous constructive suggestions.

Reviewer #2 (Remarks to the Author):

The authors addressed my concerns adequately and the manuscript would be acceptable for publication in my view.

Minor points:

Legend to Rebuttal Letter Fig. 1: CryoSPARCv3.

We have corrected this typo in Line #937. We thank the reviewer's recognition of the important biological questions addressed in this paper, and appreciate all the previous constructive suggestions.

Reviewer #3 (Remarks to the Author):

The manuscript revision now includes and is greatly improved by the low resolution structure of the substrate free PAPP-A mutant (E483A). The authors point out the difficulties in generating a structure of the wild type PAPP-A and acknowledge this in the manuscript. This addition has provided evidence to support that the unbound and bound enzyme is found as a dimer with similar dimerization mode ie trans.

The corrections add sufficient explanation of the possible mechanisms underlying the substrate recognition of IGFBP-4 versus IGFBP-5 and it is reasonable to suggest that having the complex of PAPP-A with IGF:IGFBP-4 is beyond the scope of this manuscript as that represents a significant body of work.

The addition of the fluorescence polarization assays for FL IGFBP-5 and the fragment has provided some clarity regarding the binding to PAPP-A and the authors are conservative in their conclusions with respect to the possibility of other interactions contributing to binding.

Since the original review of this manuscript a paper has been published describing the role of the linker domain in IGF-dependent protease cleavage of IGFBP-2 (Jaipuria et al Proteins DOI 10.1002/prot.26350). The conclusions are that "L-hIGFBP2 does not undergo a well-defined conformational change upon binding its ligand, but binding is accompanied by a significant change in dynamics on both the millisecond-microsecond and picosecond-nanosecond time scales". They conclude it is the change in flexibility of the linker that increases the protease susceptibility. This information should be included in the current manuscript in the discussion of possible mechanisms of PAPP-A cleavage

of IGFBP-4 and IGFBP-2. Furthermore, it would be good to acknowledge that it has not been possible to generate a structure of the IGFBP linker domain even when in complex with an IGFBP. Even in the recent cryo EM structure of IGF-I:IGFBP-3:ALS complex (Kim et al Nat Comm 2022) the linker domain remains flexible and very little density is seen for the linker domain. This information supports the suggestion that there is no significant gaining of structural stability upon ligand binding and would suggest that if there is any change in structure it is likely to be in a limited region rather than across the entire linker.

This is a great suggestion to cite to those two recent published literatures, which agree with our observation for the flexibility of IGFBP5. We further included a paragraph in the 'Discussion' session and changes could be found in Lines 357-370. We thank reviewer's recognition of the biological questions addressed in the manuscript and appreciate all the constructive feedback.